# G-protein coupled receptor 35 (GPR35) regulates the colonic epithelial cell response to enterotoxigenic *Bacteroides fragilis*

Annemarie Boleij [1,7 ✉], Payam Fathi[1], William Dalton[2], Ben Park[2,8], Xinqun Wu[1], David Huso[3,9], Jawara Allen[1], Sepideh Besharati [4], Robert A. Anders[4], Franck Housseau [5], Amanda E. Mackenzie[6], Laura Jenkins[6], Graeme. Milligan [6], Shaoguang Wu[1,10] & Cynthia L. Sears [1,10]

G protein-coupled receptor (GPR)35 is highly expressed in the gastro-intestinal tract, predominantly in colon epithelial cells (CEC), and has been associated with inflammatory bowel diseases (IBD), suggesting a role in gastrointestinal inflammation. The enterotoxigenic *Bacteroides fragilis* (ETBF) toxin (BFT) is an important virulence factor causing gut inflammation in humans and animal models. We identified that BFT signals through GPR35. Blocking GPR35 function in CECs using the GPR35 antagonist ML145, in conjunction with shRNA knock-down and CRISPRcas-mediated knock-out, resulted in reduced CEC-response to BFT as measured by E-cadherin cleavage, beta-arrestin recruitment and IL-8 secretion. Importantly, GPR35 is required for the rapid onset of ETBF-induced colitis in mouse models. GPR35-deficient mice showed reduced death and disease severity compared to wild-type C57Bl6 mice. Our data support a role for GPR35 in the CEC and mucosal response to BFT and underscore the importance of this molecule for sensing ETBF in the colon.

[1] Johns Hopkins University, Department of Medicine, Division of Infectious Diseases, Baltimore, MD, USA. [2] Johns Hopkins University, Department of Oncology Center-Hematologic Malignancies, Baltimore, MD, USA. [3] Johns Hopkins University, Department of Molecular and Comparative Pathobiology, Baltimore, MD, USA. [4] Johns Hopkins University, Department of Pathobiology, Baltimore, MD, USA. [5] Johns Hopkins University, Department of Oncology Sidney Kimmel Comprehensive Cancer Center, Baltimore, MD, USA. [6] Centre for Translational Pharmacology, Institute of Molecular, Cell, and Systems Biology, College of Medical, Veterinary and Life Sciences, University of Glasgow, Glasgow, Scotland, UK. [7] Radboud University Medical Center (Radboudumc), Department of Pathology, Radboud Institute for Molecular Life sciences (RIMLS), Nijmegen, The Netherlands. [8] Present address: Vanderbilt University Medical Center, Department of Medicine, Division of Hematology and Oncology, Nashville, Tenessee, USA. [9] Deceased: David Huso. [10] These authors contributed equally: Shaoguang Wu, Cynthia L. Sears. ✉ email: Annemarie.Boleij@radboudumc.nl

The microbiome has been demonstrated to make considerable contributions to both health and disease. Genome-wide association studies (GWAS) and microbiome sequencing have provided powerful tools to study the etiologic importance of host genetic susceptibility and unbalanced gut microbiota to promote inflammatory bowel diseases (IBD). However, understanding how these factors work together to trigger IBD remains a challenge.

Several studies have identified an association between enterotoxigenic *Bacteroides fragilis* (ETBF) that secretes the *Bacteroides fragilis* toxin (BFT) and human IBD, as well as colorectal cancer (CRC)[1–3]. ETBF is associated with acute diarrhea in children and adults, and asymptomatic colonization is likely to be frequent, at least, in some populations (~0–55%, depending on the study)[4–6]. It is likely that multiple *B. fragilis* strains can coexist in the GI tract and that competition between these strains is determined by their adherence and other virulence factors. The pathogenesis of ETBF is dependent on the secretion of BFT, a zinc-binding metalloprotease[7–9]. *B. fragilis* with a deleted *bft* gene is unable to cause inflammation in mouse colitis models[9]. Three different BFT isoforms have been identified and named as BFT-1, BFT-2, and BFT-3[10], whose encoded amino acid sequences are >93% identical[10,11]. Thus, BFT production and activity might depend on the *B. fragilis* strain and secreted BFT isotype in vivo. While *bft* expression can be inhibited by glucose and fermentable carbohydrates present in the GI tract, oxygen and the cysteine protease fragipain (Fpn) present in high concentrations in the mucus layer upregulate *bft* expression potentially enabling successful colonization in the gut mucosa[12,13].

On the mucosa, BFT directly interacts with colonic epithelial cells (CECs) triggering cell morphology changes[14], actin rearrangement[15], E-cadherin degradation[16–18], as well as activation of a spectrum of cellular signaling pathways, including NF-kB (resulting in IL-8 secretion)[19], ERK/P38 MAPKs, and Wnt/beta-catenin pathways corresponding with increased CEC expression of c-Myc[9,16–18,20]. Consistent with the capacity of BFT to diminish colon barrier function, mice colonized (by oral gavage) with ETBF develop acute IL-17-dominant colitis followed by protracted (~1 year) ongoing colonization, chronic colon inflammation, and persistent excess mucosal IL-17[21,22]. Importantly, ETBF colonization of multiple intestinal neoplasia mice (*Apc*[+/Min]), a Wnt-signaling mouse tumorigenesis model, induces IL-17- and NF-kB-dependent distal colon tumor formation[9,23–25]. These observations combined with human research data[1–3,26,27] suggest that ETBF colonization may be a risk factor for human colon carcinogenesis. Further, our recent report suggests that ETBF may act synergistically with another oncogenic bacterium, polyketide synthase-positive *Escherichia coli* (*pks+ E. coli*) to initiate tumor formation in patients with the familial adenomatous polyposis (FAP) syndrome[28]. However, many fundamental questions remain regarding how CECs sense environmental changes and interact with BFT to maintain homeostasis or promote disease upon ETBF colonization.

Recently, several reports have suggested that the G protein-coupled receptor (GPR)35 may be related to gastrointestinal inflammation and colitis. In 2009, a GWAS for early-onset IBD linked a single-nucleotide polymorphism (SNP) in GPR35 (rs4676410) to ulcerative colitis[29]. Later, a second GWAS study showed that a missense SNP within GPR35 (rs3749171) was associated with both primary sclerosing cholangitis (PSC) and ulcerative colitis[30]. More recently, another GWAS and immunoChip study reported an association between a GPR35 polymorphism at 2q37 and Crohn's disease[31]. Together, these reports suggest that GPR35 polymorphisms contribute to the pathogenesis of IBD.

GPR35 is an orphan receptor first identified in 1998 by O'Dowd et al.[32]. The protein has seven transmembrane regions and comprises 309 (short isoform) or 340 amino acids (long isoform). A connection between GPR35 signaling and cell transformation and/or proliferation has been proposed by Okumura et al. through their initial identification of the long GPR35 isoform (GPR35b)[33]. To date, five different splice variants have been described (Ensembl, ENSG00000178623). GPR35 is highly expressed in the gastrointestinal (GI) tract, from the stomach to rectum, and enhanced in CRC tissues based on data from the Human Protein Atlas[34]. It is predominantly expressed in the intestinal crypt enterocytes and in several immune cells, such as invariant natural killer T cells and peripheral blood mononuclear cells (PBMCs)[35,36]. Kynurenic acid, a metabolite of L-tryptophan, has been proposed to be the endogenous ligand of GPR35[35]. Upon kynurenic acid-mediated GPR35 activation, β-arrestin 2 translocates to the cell membrane to mediate GPR35 internalization, resulting in receptor desensitization[37]. Studies on the roles of kynurenic acid in immunity and inflammation have suggested a role for kynurenic acid–GPR35 interactions and microbiota-associated kynurenic acid metabolism in gut homeostasis. Schneditz et al. reported recently that GPR35 promotes glycolysis, proliferation, and oncogenic signaling by engaging with the sodium potassium pump in CECs[38]. This knowledge suggests that GPR35 could play an important role in host–microbiome interactions.

In this study, we identified GPR35 as crucial to BFT action in vitro and then extended these results using in vivo mouse models to test the hypothesis that GPR35 expressed on the CEC membrane senses and reacts to BFT. We demonstrate that GPR35 is an important regulator of the early response to ETBF infection.

## Results

**Identification of GPR35 as a signaling molecule for BFT.** In an attempt to identify essential CEC molecules mediating the initial cellular interaction with BFT, a microarray data subtraction comparison among the BFT-responsive CRC epithelial cell lines HT29/c1 (in this study), HT29 (GSM396550), T84 (in this study), Caco-2 (GSM24832), and SW480 (in this study) versus the BFT nonresponsive kidney cell line HEK293 (in this study) and HeLa cells (GSM410912) was conducted. The total RNA was extracted using Direct-zol RNA kit (ZYMO research) from cultured cells, and the microarray was performed using Human transcriptome array 2.0 (Affymetrix) by the JHMI Deep Sequencing & Microarray Core facility. We identified 82 epithelial membrane-related proteins as potential receptor candidates in BFT-responsive cells (Supplementary Data 1). We used lentiviral shRNA knockdown (KD) screening of these 82 proteins to identify candidate proteins that diminished or blocked the morphological changes in HT29/c1 cells (details on shRNA infection below) induced by BFT treatment (5 nM) for 3 and 24 h. Based on this initial screening, three clones emerged as they showed reduced morphological change [c1–8 (GPR35, TRCN0000008887), c6–9 (Claudin-4, TRCN0000116627), and c3–7 (RAB20, TRCN0000048093)] (Supplementary Table 1). These three clones and the membrane receptor CD44 (TRCN0000057563) as control were selected to generate stable cell lines. Morphology assay and E-cadherin western blot were performed with and without BFT for 1–3 h to confirm the loss of BFT activity in these stable cell line clones.

The stable KD clone 1–8 targeting the 3UTR region of GPR35 (*GPR35* mRNA reduced to 79% of untreated control, $N = 3$, $P < 0.01$, Fig. 1a) showed a nearly complete inhibition of BFT biological activity based on morphology change (Fig. 1b), reduced IL-8 secretion (Fig. 1c), and reduced E-cadherin cleavage

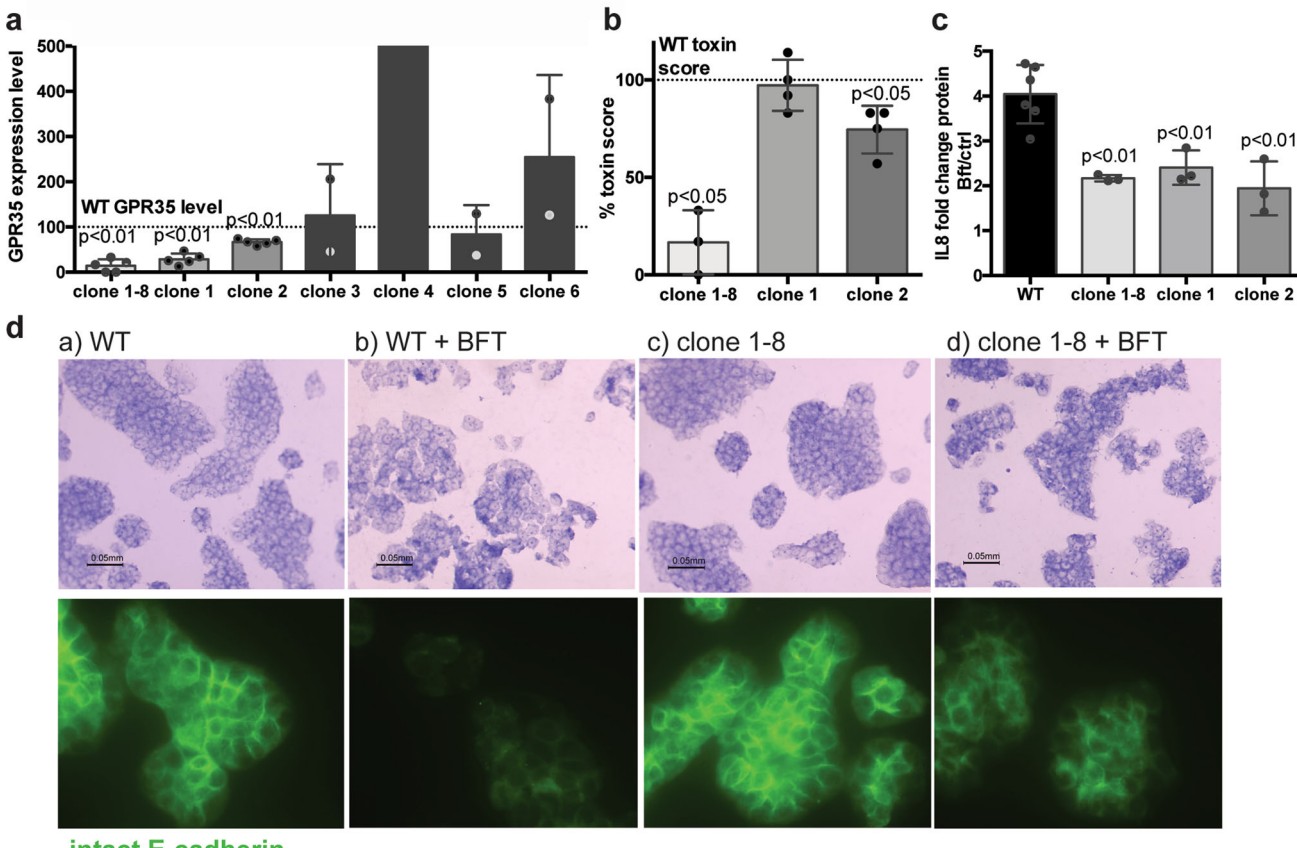

**Fig. 1 Clone 1–8 GPR35 KD by shRNA results in reduced morphology change, E-cadherin cleavage, and IL-8 secretion. a** GPR35 mRNA expression levels relative to GAPDH. GPR35 expression is significantly reduced compared to wild-type (WT) HT29/c1 cells ($N = 8$) in the GPR35 shRNA clones 1–8 ($N = 5$, $P < 0.01$), 1 ($N = 5$, $P < 0.01$), and 2 ($N = 5$, $P < 0.01$), but not in clones 3, 4, 5, and 6 ($N = 2$ each, one-sample $T$ test). Dashed line represents WT GPR35 level. Two points of clone 4 (540 and 947%) outside the $y$ axis. **b** Morphology changes of HT29/c1 cells were reduced by clone 1–8 ($N = 3$, $P < 0.05$), and clone 2 ($N = 4$, $P < 0.05$), but not clone 1. Dashed line represents WT toxin score (100%). **c** IL-8 secretion was significantly reduced in all three GPR35 shRNA clones (clone 1–8, clone 1, and clone 2) ($N = 3$, $P < 0.01$) compared to HT29/c1 WT cells ($N = 6$). **d** Representative pictures of morphology (top panels) and intact E-cadherin (green, bottom panels). Untreated HT29/c1 WT and GPR35 shRNA clone 1–8 cells are shown in (a) and (c), respectively. After 3 h, BFT changed morphology and cleaved E-cadherin in WT cells (b), but not in the GPR35 shRNA clone 1–8 cells (d).

compared to the vector control by both immunofluorescence and western blot (Fig. 1d, Supplementary Figs. 1a and 5b). None of the other shRNA clones, targeting Claudin-4, CD44, and RAB20, inhibited E-cadherin cleavage by BFT (Supplementary Figs. 1a and 5b). Thus, out of these four proteins, only GPR35 emerged as a candidate for BFT signaling on the CEC membrane.

Next, we generated additional stable GPR35 shRNA KD clones from HT29/c1 cells infected with 6 different shRNA lentiviral constructs (see "Methods"). GPR35 mRNA was significantly reduced in two construct-infected cells named clone 1 ($-71.3\%$, $N = 5$, $P < 0.01$), and clone 2 ($-33.4\%$, $N = 5$, $P < 0.01$); no other clones demonstrated GPR35 KD (clones 3–6, $N = 2$ each, Fig. 1a). Therefore, we further evaluated both clone 1 and clone 2. While the original clone 1–8 almost completely blocked BFT-induced morphological change (Fig. 1b), the newer GPR35 KD clones 1 and 2 revealed only a modest reduction in BFT-induced HT29/c1 cell morphology changes (clone 1, no effect; clone 2, 25.5% reduced cell morphology score, $N = 4$, $P < 0.05$) (Fig. 1b) and no blockade of E-cadherin cleavage. When analyzing downstream IL-8 signaling, all GPR35 KDs (clones 1–8, clones 1 and 2) displayed a similar decrease in IL-8 protein secretion ($2.17 \pm 0.04$ (clone 1–8), $2.41 \pm 0.23$ (clone 1), $1.95 \pm 0.35$ (clone 2), vs $4.04 \pm 0.27$ (wild type), $N = 3$, $P < 0.01$, Fig. 1c). Collectively, our shRNA KD results suggest a role for GPR35 in BFT signaling although this likely depends on the level of KD and/or the remaining

protein function and expression. Because high-affinity, specific GPR35 antibodies are currently unavailable, we had to rely on mRNA to analyze GPR35 expression levels without the ability to accurately assess protein levels. We further note that even in clone 1–8 that displayed the highest KD-level and the corresponding downstream effects, cell response following BFT treatment was only temporarily delayed (~3 h) but not completely blocked by the GPR35 KD.

**GPR35 expression in CEC lines correlates with cell sensitivity to BFT.** To further evaluate the relationship between GPR35 expression and the cell response to BFT, we investigated the mRNA level of GPR35 isoform a (GPR35a, 309 amino acids) and isoform b (GPR35b, 340 amino acids) in several CEC lines. The long isoform (GPR35b) has 31 extra amino acids at the N-terminal site of the protein (Supplementary Fig. 1b). Our results suggest that GPR35b is the dominant isoform present in the colon carcinoma cell lines examined, matching data in public databases (Supplementary Fig. 1c). When normalized to GAPDH expression, HT29/c1 cells have the highest mRNA level of GPR35 followed by Caco-2 cells, whereas HCT116 cells showed the lowest GPR35 expression levels. Consistent with these data, HCT116 cells display limited cell morphologic changes and E-cadherin cleavage after BFT treatment when compared to HT29/c1 cells. As an additional assessment, IL-8 expression after BFT treatment was more pronounced in HT29/c1

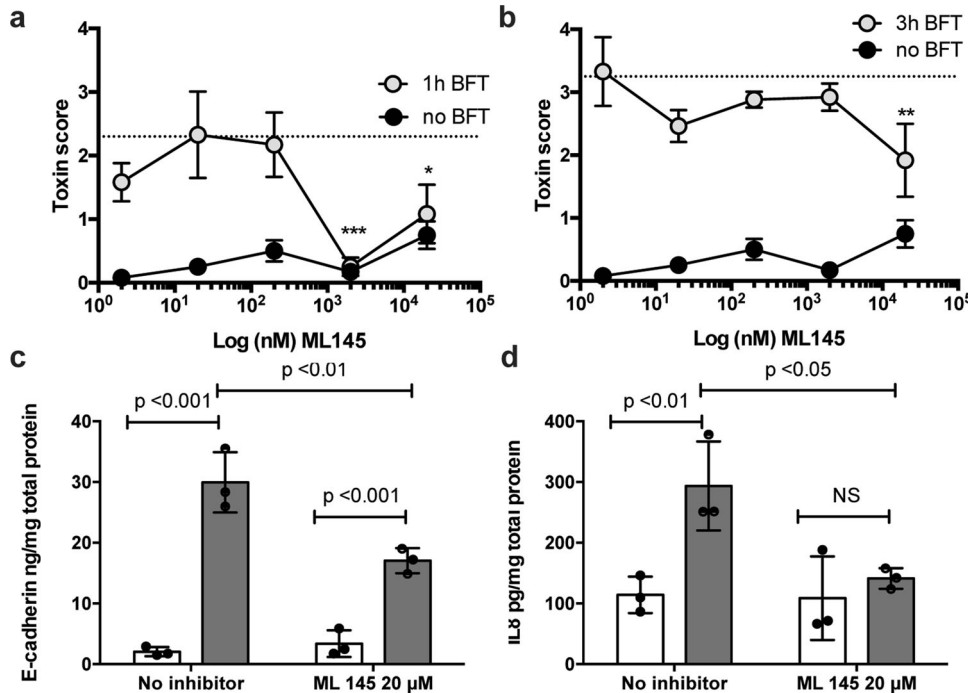

**Fig. 2 The GPR35 antagonist ML145 inhibits CEC response to BFT in HT29/c1 cells.** HT29/c1 cells were treated for 1 h (**a**) and 3 h (**b**) with BFT and increasing concentrations of the GPR35 antagonist ML145 ($N = 3$). Treatment with ML145 significantly inhibited changes in HT29/c1 cell morphology (two-way ANOVA $P = 0.023$ and $P = 0.015$, 1 h (**a**) and 3 h (**b**), respectively) compared to the positive control (BFT without antagonist—dotted lines). Treatment with 2 and 20 μM of ML145 inhibited BFT morphological changes at 1 h and 20 μM ML145 at 3 h (Dunnet's multiple comparison test *$P < 0.05$, **$P < 0.01$, ***$P < 0.001$). **c** BFT treatment significantly enhanced the release of E-cadherin into the culture supernatant ($P < 0.001$). E-cadherin release was significantly inhibited by ML145 ($N = 3$, $P < 0.01$). **d** BFT treatment significantly enhanced IL-8 secretion into the culture supernatant ($P < 0.01$). IL-8 secretion was significantly inhibited by ML145 ($N = 3$, $P < 0.05$; two-way ANOVA with Bonferroni post test in **c** and **d**).

than in HCT116 cells (Supplementary Fig. 1d; 9.46-fold and 3.34-fold, respectively, $P < 0.01$, $N = 2$). These data suggest a correlation between the level of GPR35 expression and cell sensitivity to BFT.

**BFT signaling is inhibited by the selective GPR35 antagonist ML145.** The chemical antagonist of GPR35, ML145, has been shown to specifically inhibit GPR35 signaling by the known agonists Zaprinast (PubChem CID 135399235) and compound 10 (PubChem CID 2295463)[39,40]. ML145 has a high affinity for human GPR35 and, for example, has a >1000-fold higher affinity toward GPR35 than GPR55; ML145 is the most selective known antagonist for GPR35[41]. To further investigate whether GPR35 is involved in BFT signaling, we pretreated HT29/c1 cells (30 min) with 20 μM ML145 followed by incubation with BFT to determine morphological changes, E-cadherin cleavage, and IL-8 secretion. BFT-induced morphological changes were significantly reduced for up to 3 h by treatment with the inhibitor ML145 ($1.92 \pm 1.01$, cell morphology score, see "Methods") when compared to treatment with BFT alone ($3.17 \pm 0.14$ two-way ANOVA $P = 0.023$ (1 h) and $P = 0.015$ (3 h) $N = 3$) (Fig. 2a, b). Reduced cell morphological change by ML145 correlated with a decrease in the release of the 80-kDa extracellular domain of E-cadherin from HT29/c1 cells when compared to cells treated with BFT alone (3.2-fold vs 12.2-fold ($P < 0.01$, $N = 3$), respectively) (Fig. 2c). IL-8 secretion by HT29/c1 cells induced upon BFT treatment was also inhibited by ML145 at 20 μM (139.5 pg/ml ML145+BFT vs 302.5 ng/ml BFT alone, $P < 0.05$, $N = 3$, Fig. 2d). Together, these data further support a role for GPR35 in facilitating BFT-induced HT29/c1 cell morphologic changes, E-cadherin release, as well as increased secretion of IL-8.

**BFT stimulates β-arrestin recruitment in HT29/c1 cells supporting GPCR involvement.** G protein-coupled receptor (GPCR) signaling involves the recruitment of beta-arrestins (β-arr) to desensitize and internalize GPCRs[42]. Recent research indicates that β-arrs can also function as signal transducers for GPCRs and result in activation of MAPK and ERK pathways via interaction with Src family kinases[43]. For GPR35, β-arr2 is considered the most substantially involved in response to agonists[44]. To evaluate whether GPCR signaling is involved in the cellular response to BFT, β-arr recruitment was measured in HT29/c1 cells exposed to BFT or compound 10, a known GPR35 agonist. HT29/c1 cells stimulated with BFT had an increased intracellular β-arr2 signal measured at 30 and 60 min ($P < 0.001$, $N = 6$) similar to the positive control (GPR35 agonist compound 10) (Fig. 3a, b). We noted that BFT stimulation for 4 h (240 min) increased the β-arr1 and β-arr2 signal in the membrane-associated region when compared to nonstimulated cells (Supplementary Fig. 2a; $N = 3$). An increased perinuclear staining was also noted for β-arr2 following BFT treatment. Analysis of cytoplasmic and membrane-associated proteins from HT29/c1 cells stimulated with BFT for 4 h using western blotting confirmed increased membrane-associated signals for both β-arr1 and β-arr2 (Supplementary Fig. 2b, Supplementary Fig. 5c–f). Together, these experiments indicate that exposure of HT29/c1 cells to BFT activates β-arr1 and β-arr2 signaling consistent with our hypothesis that BFT induces GPR35 signaling.

Stable KD clones of β-arr1 and β-arr2 were generated using lentiviral shRNA in HT29/c1 cells to determine whether the presence of β-arr1 and/or β-arr2 is required for BFT-stimulated GPCR signaling (Supplementary Fig. 2c). BFT stimulation of the β-arr1 and β-arr2 KD clones had no effect on morphology

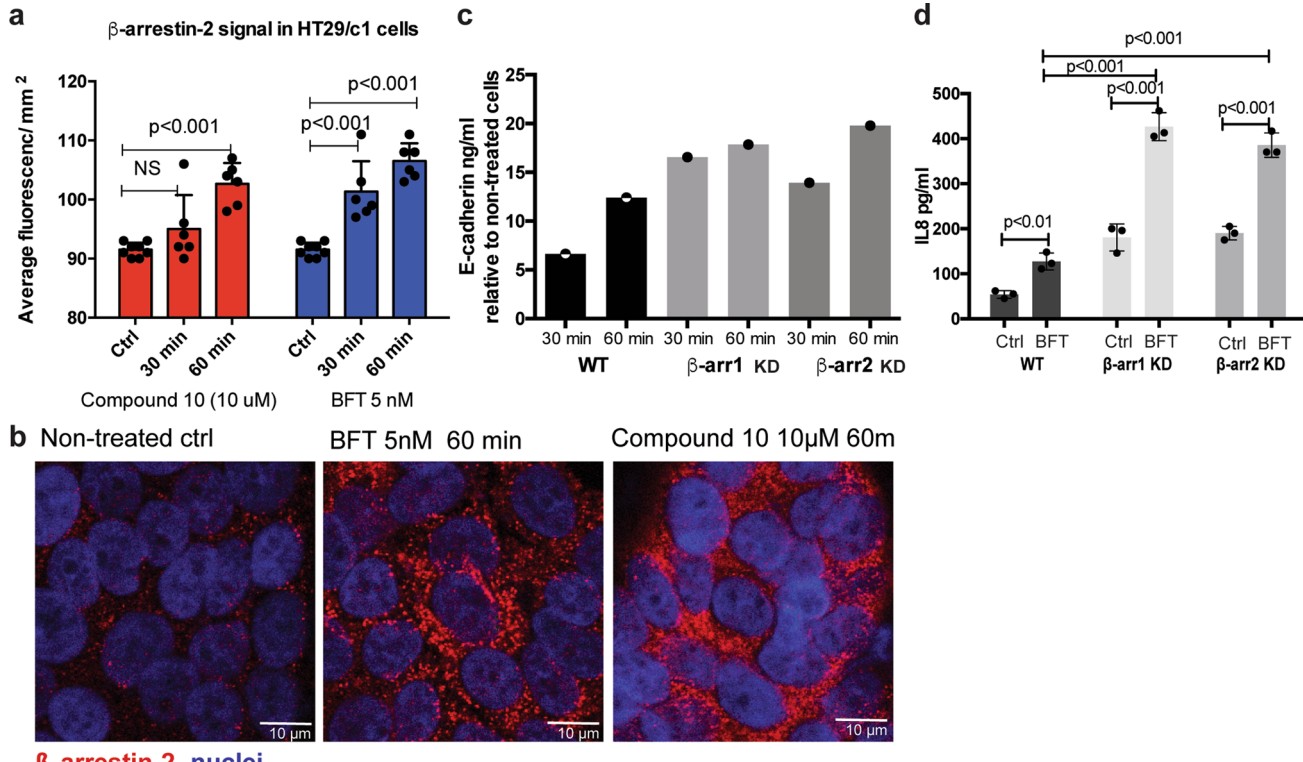

**Fig. 3 β-arrestin signaling following BFT treatment. a** The β-arr2 signal in HT29/c1 cells was increased after 30 and 60 min of treatment with BFT to levels similar to that of cells treated with the GPR35 agonist compound 10 ($P < 0.001$, $N = 6$). Representative images of β-arr2 staining are shown in (**b**) at ×400 magnification. **c** E-cadherin cleavage with the release of the E-cadherin extracellular domain was enhanced in β-arr1 and −2 KD cells after both 30 and 60 min of BFT treatment compared to control BFT-treated HT29/c1 cells ($N = 1$). **d** Baseline IL-8 secretion was increased in β-arr1 and −2 KD cells compared to control cells ($P < 0.001$, $N = 2$ each). While IL-8 secretion after BFT treatment was increased in the β-arr1 and β-arr2 KD ($P < 0.001$), the overall fold change in IL-8 secretion (BFT vs nontreated ctrl) was similar in β-arr1 amd β-arr2 KD cells compared to control cells.

change, demonstrating that β-arr1 and β-arr2 are not required for BFT changes in cellular shape. Rather, the BFT-induced release of the extracellular domain of E-cadherin was enhanced in the stable β-arr KD clones when compared to control cells (6.63-fold increased release wild type, 16.6-fold β-arr1 KD and 13.9-fold β-arr2 KD, $N = 1$) (Fig. 3c), while IL-8 secretion was similarly induced in wild-type and KD cells following 4 h of BFT stimulation (2.41-fold increase wild type ($N = 3$), 2.43-fold β-arr1 KD ($N = 3$), and 2.03-fold β-arr2 KD ($N = 3$); one-way ANOVA $P = 0.64$). β-arr1 and β-arr2 KD also resulted in a significant baseline increase ($P < 0.001$) in IL-8 secretion (Fig. 3d). The observed enhancement of CEC E-cadherin release and IL-8 secretion following BFT treatment in the β-arr1 and β-arr2 KD cells supports the desensitizing role played by β-arrestins following BFT treatment.

**CRISPRcas GPR35 KO in HT29/c1 cells also results in delayed BFT response.** To further confirm the function of GPR35 during the CEC response to BFT stimulation, we generated three sequence-confirmed GPR35 KO clones in HT29/c1 cells with the CRISPRcas system using four guide RNAs (gRNAs) located on the first extracellular and transmembrane domains (amino acids 10–43) of GPR35a. Clone 2AH5 resulted in 39% frameshift deletions ranging from 4 to 29 bp that result in alternative protein coding around Tyr 26, and 3% resulted in in-frame deletions removing Val 32 (3 bp) or Leu 31-Leu 39 (21 bp) that retained high similarity to wild-type GPR35 protein (Supplementary Fig. 3a, b). Clone 2BE5 showed an 8-bp GPR35 deletion (36%) that resulted in a frameshift translation starting at Leu 40

(GPR35a); however, 32% of the sequences resulted in an in-frame deletion of 9bp, resulting in deletion of 3 amino acids at Leu 40 (19%) or Asn 38 (13%) while again retaining high similarity to the wild-type GPR35 protein. In clone 2AA9, only heterogenous deletions were observed consisting of 6% in-frame and 6% frameshift deletions at Tyr 26 ranging from 9 to 28 bp (Supplementary Fig. 3a, b). Protein alignment using CLUSTALW Omega and a phylogenetic tree based on a point mutation matrix (PAM) of the suggested protein translations of the GPR35 KO clones showed high similarity between KO 2AA9, 2BE5, and wild-type GPR35, but not 2AH5 (Supplementary Fig. 3a, b).

Real-time PCR that amplified codon regions Ile (156) to Ala (185) showed that GPR35 mRNA was reduced 49% for clone 2AH5 ($N = 4$, $P < 0.05$), 79% for clone 2AA9 ($N = 4$, $P < 0.01$), and 48% for clone 2BE5 ($N = 2$, NS) compared to wild-type HT29/c1 cells (Fig. 4a). Only clone 2AH5 displayed a significant decrease in HT29/c1 cell morphology change after BFT treatment (decrease 92.2% for 2AH5 ($N = 4$, $P < 0.01$), while clone 2BE5 showed a heterogeneous decrease in cell morphology change (39.2%, $N = 4$, NS) and no change was observed for clone 2AA9 ($N = 4$, NS) (Fig. 4b, c)). This corresponded with a reduction in E-cadherin cleavage of 22.4% for clone 2AH5 compared to the BFT control, but little (7.5% decrease) to no impact on E-cadherin cleavage for clone 2AA9 and clone 2BE5, respectively (Fig. 4d and Supplementary Fig. 5a). IL-8 secretion was reduced only in clone 2AH5 ($N = 2$, $P < 0.05$), but was not changed in 2BE5 ($N = 2$, NS), while clone 2AA9 showed significantly increased levels of IL-8 ($N = 2$, $P < 0.05$) (Fig. 4e).

Although the previous experiments suggested a reduced expression of GPR35 for clones 2AA9 and 2BE5 (Fig. 4a), no

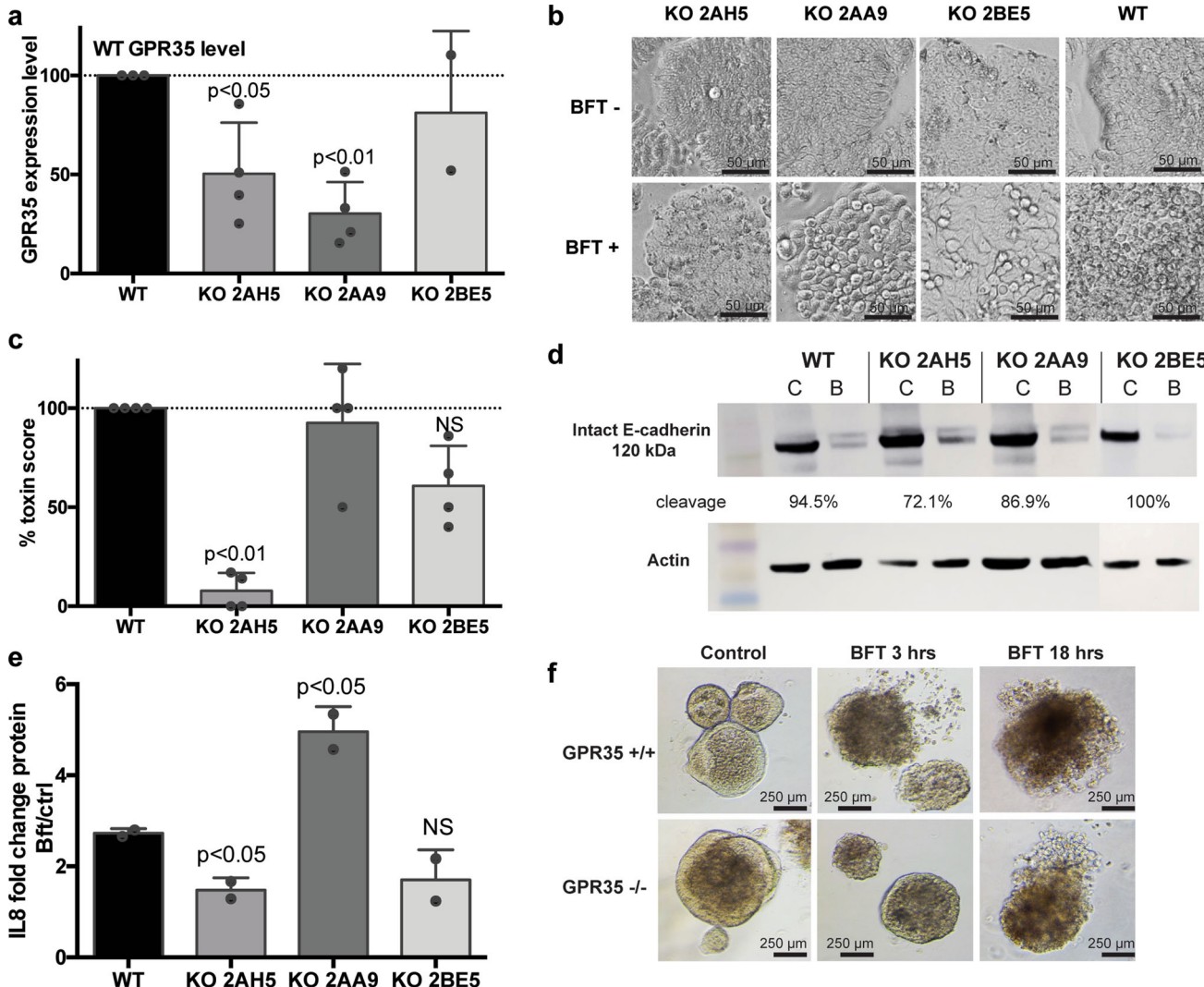

**Fig. 4 GPR35 knockout in HT29/c1 cells and mouse organoids results in delayed BFT response. a** CRISPRcas KO of GPR35 in three confirmed clones 2AH5, 2AA9, and 2BE5 resulted in reduced GPR35 mRNA expression levels for 2AH5 ($N = 4$, $P < 0.05$) and 2AA9 ($N = 4$, $P < 0.01$) but not for 2BE5 ($N = 2$). **b**, **c** Morphology changes following BFT treatment in CRISPRcas KO were only reduced for clone 2AH5 but not for 2AA9 and 2BE5. **d** E-cadherin cleavage, determined through western blotting, was reduced to 72% in 2AH5 compared to 94.5% in WT HT29/c1 cells, but was not totally inhibited. A representative western blot and corresponding actin controls are depicted ($N = 3$). **e** The fold change of IL-8 secretion in BFT-treated versus nontreated control cells was significantly reduced only for the CRISPRcas KO cell line 2AH5 ($N = 2$, $P < 0.05$). **f** Enteroids derived from the WT and GPR35$^{-/-}$ distal mice colon exposed to BFT show a delayed change in morphology at 3h after treatment with BFT in the GPR35$^{-/-}$ condition.

significant alterations in morphological change, E-cadherin cleavage, and/or IL-8 secretion were detected, suggesting that BFT signaling was maintained in these clones, either due to their high similarity to wild-type GPR35 resulting from the in-frame deletion (2BE5) or because of a low percentage of mutant DNA in the sample (2AA9). It is likely that the high similarity in sequence between the clones 2AA9, 2BE5, and wild-type GPR35 resulted in the preservation of BFT signaling. This is in contrast to the reduced BFT-dependent signaling observed in clone 2AH5. Importantly, both CRISPRcas clone 2AH5 and shRNA clone 1–8 only yielded a short-term delay in the CEC response to BFT, suggesting that while GPR35 contributes to BFT signal transduction, inhibition of this pathway alone is insufficient to abrogate CEC responsiveness to BFT.

To further investigate the role of GPR35 in CECs, enteroids derived from the colons of wild-type -C57Bl6 and GPR35$^{-/-}$ (KO) mice were exposed to BFT (25 nM) for 3 h. GPR35$^{-/-}$ enteroids also showed a delayed morphological change following BFT treatment when compared to wild-type -C57Bl6 enteroids

($N = 3$, Fig. 4e). The ex vivo enteroid data recapitulate the delayed responsiveness to BFT observed with ML145 GPR35 inhibition, shRNA KD (clone 1–8), and KO by CRISPRcas (clone 2AH5). Together, these data suggest that colon CEC GPR35 functions, in part, as a signaling molecule for the CEC response to BFT, but loss of GPR35 is insufficient to prevent in vitro BFT–CEC activity and signaling.

**GPR35 does not facilitate BFT binding to the intestinal epithelial cells.** These in vitro results point toward an important role for GPR35 in BFT-induced cellular signaling. In previous research, we have shown that BFT interacts with an epithelial cell receptor on colon HT29/c1 cells[20]. To investigate whether GPR35 is a binding partner for BFT, recombinant BFT labeled with Alexa 488 was visualized using confocal microscopy in HT29/c1 wild-type cells and shRNA KD clones 1–8, 1 and 2. Although morphological change was significantly reduced in clone 1–8, BFT bound similarly on the epithelial cell surface of all KD clones and

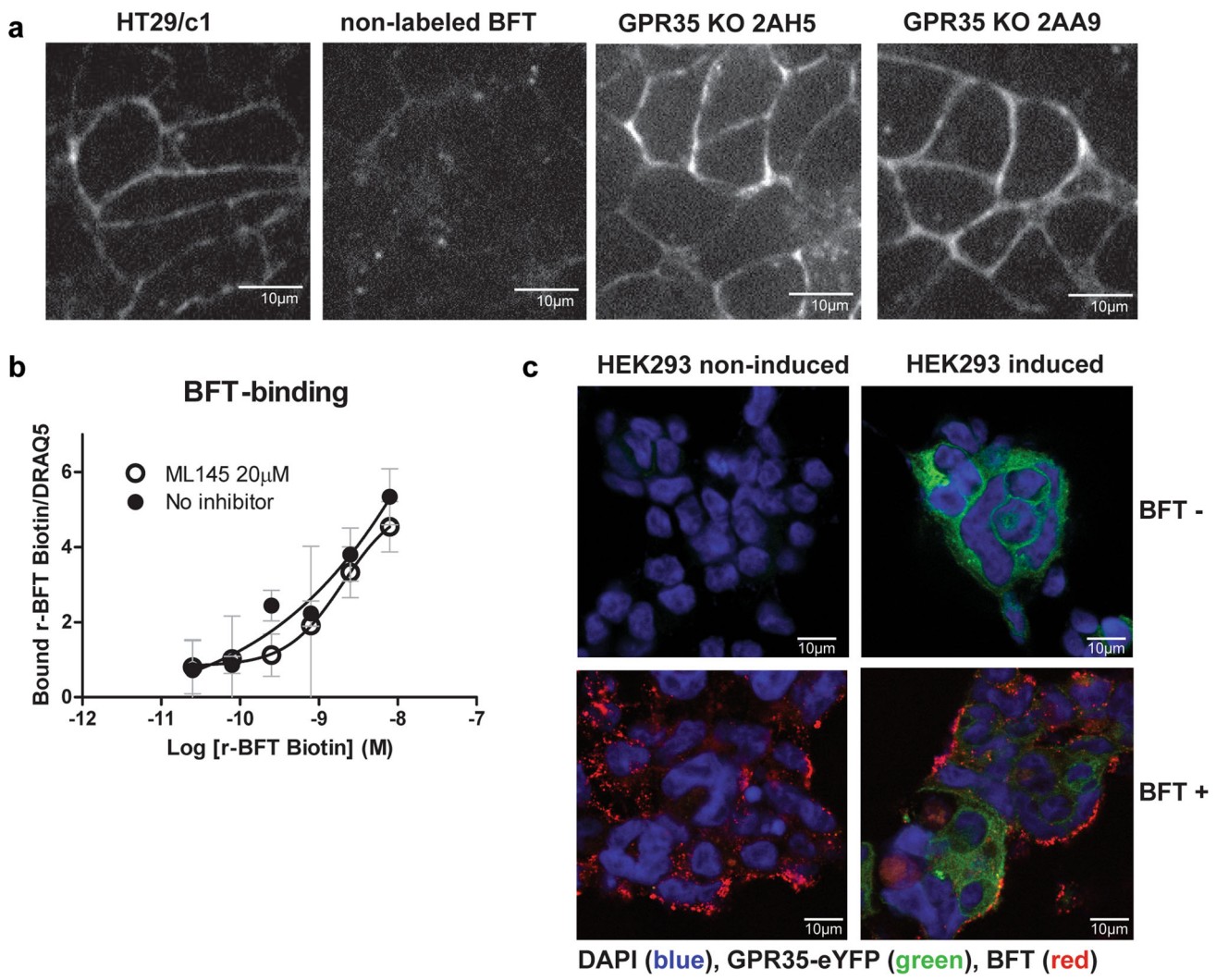

**Fig. 5 GPR35 does not facilitate BFT binding to CECs. a** HT29/c1 WT, GPR35 KO 2AH5, and GPR35 KO 2AA9 cells incubated with alexa488-labeled BFT. Blocking of 488-labeled binding with non-labeled BFT was used to show the specificity of the BFT–CEC binding. Images at ×600 magnification. **b** Binding of concentration series of biotin-labeled recombinant BFT to HT29/c1 epithelial cells and cells treated with 20 μM ML145 (NS). **c** BFT binding to HEK293 cells noninduced or induced to express human GPR35-eYFP. The nucleus is stained with DAPI (blue), BFT (red), and GPR35-eYFP (green). Images at ×400 magnification.

wild-type HT29/c1 cells. In addition, BFT binding was maintained in the 2AH5 CRISPRcas clone compared to wild-type HT29/c1 cells and the KO 2AA9 clone that served as an additional positive control (Fig. 5a). Next, we visualized biotin-labeled recombinant BFT at the epithelial cell surface in HT29/c1 wild-type cells with and without 20 μM ML145 antagonist that binds to GPR35, to further assess the direct binding of BFT to GPR35. Inhibition with ML145 did not result in a reduced binding of BFT to the epithelial cells (Fig. 5b). Similarly, doxycycline-induced overexpression of hGPR35-eYFP in HEK293 cells that do not respond morphologically to BFT did not result in morphologic changes upon BFT treatment or enhanced BFT binding to the cell surface of the HEK293 cells (Fig. 5c). Hence, while we show that GPR35 is an important interaction partner for BFT, our results do not suggest that BFT is a direct ligand of GPR35.

**GPR35 deficiency decreases the lethal reaction of mice to ETBF infection.** We next evaluated the contribution of GPR35 to the pathogenesis of ETBF in vivo using two mouse colitis models pretreated with antibiotics. Pretreatment with oral clindamycin/streptomycin (clin/strep) facilitates ETBF colonization in mice. In

the clin/strep model, mice generally survive ETBF colonization and the severity of the colitis wanes after 5–7 days post- colonization but occasional mouse death occurs. In contrast, pretreatment with oral cefoxitin, a broad-spectrum antibiotic, increases the severity of ETBF colitis and can cause high mortality during the acute phase of ETBF colitis, likely due to the near-complete elimination of the murine microbiota[45]. Herein, given our in vitro results on the early contribution of GPR35 in BFT signaling, we focused on the evaluation of the early phase of ETBF colitis. First, in an experiment with severe colitis in clin/strep pretreated mice, ETBF colonization resulted in the death of 5/6 wild-type (GPR35$^{+/+}$) mice, while none of the GPR35$^{-/-}$ mice died in the same experiment after ETBF colonization ($n = 6$ each, $P = 0.0186$, log-rank test) (Fig. 6a). In the cefoxitin-pretreated ETBF-colonized mice, 7 of 15 wild-type GPR35$^{+/+}$ mice (46.67%) and 12 of 27 GPR35$^{+/-}$ mice (44.44%) died within 7 days of ETBF colonization, whereas only 1 of 25 GPR35$^{-/-}$ mice died (4.0%) ($n = 3$ experiments, $P = 0.0367$ log-rank test) (Fig. 6b). Clinically, cefoxitin-pretreated GPR35$^{-/-}$ mice appeared less ill and gained body weight similarly to sham mice ($P = 0.5904$), differing from ETBF-colonized GPR35$^{+/-}$ or wild-type GPR35$^{+/+}$ mice that

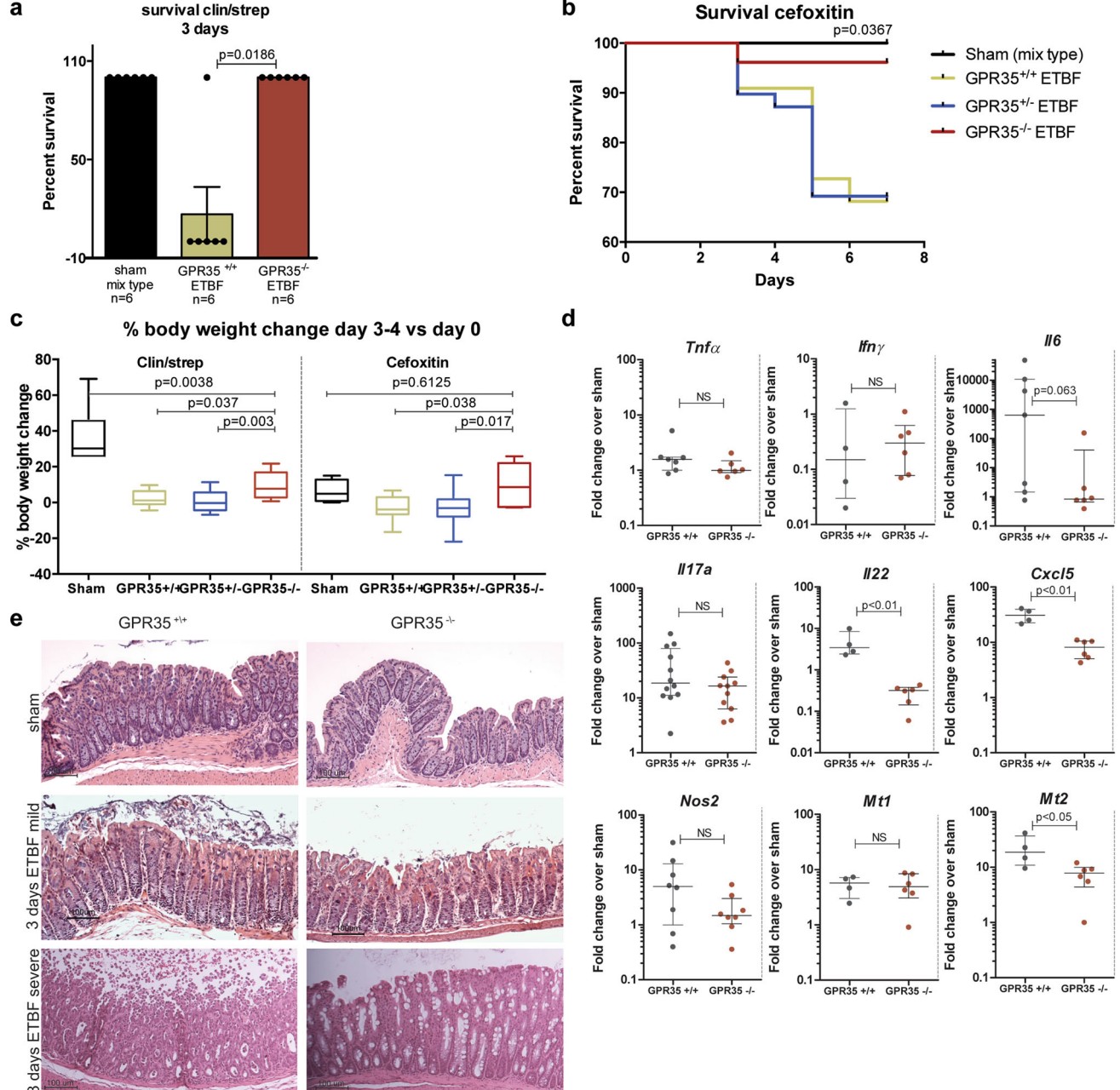

**Fig. 6 GPR35-knockout mice show reduced lethality and less severe colitis after ETBF colonization. a** Percentage survival of clin/strep pretreated GPR35$^{+/+}$ (WT) and GPR35$^{-/-}$ mice in a severe colitis experiment 3 days post ETBF infection ($P = 0.0186$) ($N = 6$ each). **b** Percentage survival of cefoxitin-treated GPR35$^{+/+}$ (WT) ($N = 22$), GPR35$^{+/-}$ ($N = 39$), and GPR35$^{-/-}$ ($N = 26$) mice over 7 days post ETBF colonization ($P = 0.0367$). **c** Mouse body weight change in cefoxitin and clin/strep-treated mice. Body weights of GPR35$^{+/+}$ (WT), GPR35$^{+/-}$, and GPR35$^{-/-}$ mice were compared 3 days after ETBF colonization to their own baseline (day 0). Sham ($N = 6$ and $N = 4$), GPR35$^{+/+}$ ETBF ($N = 9$ and $N = 7$), GPR35$^{+/-}$ ETBF ($N = 22$ and $N = 15$), and GPR35$^{+/-}$ ETBF ($N = 7$ and $N = 6$) for Clin/Strep and Cefoxitin models, respectively. **d**) *Il17a, Nos2, Il22, Cxcl5, Il6, Ifn-γ, Tnf-α, Mt1*, and *Mt2* expression in the distal colon of GPR35$^{+/+}$ and GPR35$^{-/-}$ ETBF-colonized mice compared to their sham controls at 3 days post colonization. Significant differences were observed for *Il22, Cxcl5*, and *Mt2*. Individual mice are represented in each dot. **e** H&E-stained sections of mouse distal colons from clin/strep pretreated GPR35$^{+/+}$ and GPR35$^{-/-}$ mice at 3 days post ETBF colonization compared to the sham. Mild and severe ETBF colitis histology are shown in GPR35$^{+/+}$ mice with corresponding GPR35$^{-/-}$ mice.

showed significant body weight loss ($P = 0.0452$ and 0.0158, respectively, Fig. 6c, right panel). A similar but less pronounced body weight change was observed in ETBF-colonized clin/strep-treated mice (Fig. 6c, left panel). Analysis of cytokine expression by qPCR 3 days after ETBF colonization revealed limited differences between GPR35$^{-/-}$ and wild-type GPR35$^{+/+}$ mice in the distal colon. Notably, expression of *Il17a*, a signature inflammatory cytokine for ETBF infection, and *Nos2*, a general

inflammation marker, did not differ (Fig. 6d). However, mRNA expression of *Il-22* (important for host defense and mucosa regeneration), *Cxcl5* (also known as epithelial-derived neutrophil-activating peptide 78 (ENA-78)), and metallothionein (*Mt2*) (a modulator of the immune response by zinc sequestration) were significantly reduced in GPR35$^{-/-}$ mice when compared to wild-type-infected ETBF mice. Histologically, the GPR35$^{+/+}$ mice that survived 3 days of ETBF infection showed mucosal damage,

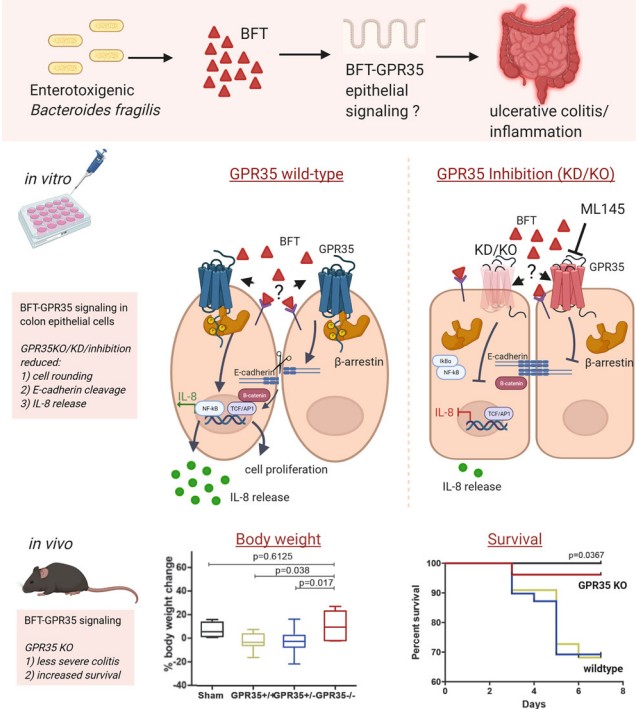

**Fig. 7 Graphical abstract.** Graphical abstract summarizing the in vitro and in vivo findings on enterotoxigenic *Bacteroides fragilis* (ETBF) toxin (BFT) and epithelial cell signaling through GPR35. In the top panel, the in vitro findings with GPR35 shRNA knockdown (KD), inhibition with GPR35 antagonist ML145, and CRISPRCas knockout (KO) on morphological change, E-cadherin cleavage, and IL-8 secretion upon stimulation of HT29/c1 cells with *Bacteroides fragilis* toxin (BFT). In the bottom panel, the in vivo findings in C57Bl6 wild-type (WT) and GPR35−/− mice pretreated with cefoxitin antibiotics on body weight change and survival after colonization with ETBF. *Created with BioRender.com.

including bleeding, epithelial cell shedding, and ulcerations. Overall, there was neither a clear difference in lamina propria immune cell infiltration, number of ulcerations, and lymphocyte aggregates between GPR35−/− and GPR35+/+ mice that survived after ETBF colonization (Fig. 6e) nor a difference in E-cadherin detection (Supplementary Fig. 4). However, CEC shedding in severe ETBF colitis in GPR35+/+ mice exceeded that observed in ETBF-colonized GPR35−/− mice (Fig. 6e, lower panel). Altogether, these results revealed decreased overall mortality, and decreased colitis-induced weight loss, and suggest that GPR35 contributes, at least, in part to early ETBF-induced colon inflammation. These in vivo results add to our in vitro results indicating that GPR35 mediates, in part, the early signaling response to BFT, and has an early impact on ETBF colitis.

## Discussion
In this study, we show that GPR35 plays an important role in the CEC response to BFT produced by ETBF (Fig. 7). Our in vitro results indicate that blocking GPR35 activity using selective antagonists of GPR35, shRNA knockdown, and CRISPRcas knockout contributes to a delayed CEC response to BFT. In addition, our ex vivo results using colon enteroids of GPR35-knockout mice confirm that GPR35 is important for sensing BFT in CECs. In two mouse models of ETBF colitis, GPR35 knockout resulted in less body weight loss, less severe colitis, and an increased survival rate at 3–7 days post ETBF colonization, despite no decrease in *Il17a* expression, a key mediator of ETBF

colitis[9]. Together, these results indicate that GPR35 plays an important role in the early CEC and mucosal response to BFT in vitro and in vivo.

HT29/c1 cells, the subclone of the human CRC cell line HT29, are highly sensitive to BFT and have been previously used as an in vitro model to study BFT–CEC interactions[14]. In addition, HT29 cells have been used as a model to screen for chemical agonists of GPR35[46,47]. In this study, we verified GPR35 expression in HT29/c1 cells[38] and showed a correspondence of GPR35 expression with the level of BFT-induced E-cadherin cleavage and IL-8 secretion in CECs. BFT treatment of HT29/c1 cells resulted in βarr1 and βarr2 recruitment, indicating activation of GPCR signaling by BFT. Cellular knockdown of βarr1 and βarr2 resulted in enhanced BFT-induced E-cadherin cleavage and IL-8 secretion, indicating that beta-arrestin recruitment is most likely involved in internalization and desensitization of GPR35, limiting to some extent BFT action[48]. For some GPCRs the β-arr-mediated signaling relies on either β-arr1 or β-arr2, and is inhibited by the other respective βarr forms (reciprocal regulation). Alternatively, GPCR signaling can also depend on both βarrs[49]. Our data suggest that both βarrs are activated upon BFT treatment of CECs.

Inhibition of GPR35 function using the selective antagonist ML145 reduced BFT-mediated changes in cellular morphology and downstream effects following BFT exposure. BFT causes actin cytoskeleton rearrangements in HT29/c1 cells similar to what occurs following exposure to the GPR35 agonists pamoic acid and zaprinast in human saphenous vein cells[50]. This morphological change in saphenous vein cells by GPR35 agonists was also inhibited with the GPR35 antagonist ML145 in line with the inhibition of BFT action[15,50]. Furthermore, a recent study investigating the signaling network of GPR35 in HT29 cells shows that stimulation of GPR35 with agonists pamoic acid and YE210 is related to changes in morphological processes, such as cytoskeletal remodeling and has a function in inflammation[47], both of which are also affected by BFT. When GPR35 mRNA was inhibited by shRNA (clone 1–8), and functionally knocked out by CRISPRcas (clone 2AH5), HT29/c1 cells displayed a significant delay in BFT response. Unfortunately, we were unable to find a functional GPR35 antibody for verification of GPR35 expression at the protein level after knockdown and knockout, and we had to rely on mRNA expression patterns and in silico modeling of GPR35 protein knockout. The latter suggested that the 2AH5-knockout clone resulted in a frameshift deletion and a different protein composition matching our functional BFT activation experiments. The KD and KO results were confirmed in GPR35-knockout mouse enteroids that demonstrated a delayed reaction to BFT, but the response was not completely eliminated. Together, our data show that blocking GPR35 impairs but does not eliminate the biological activity of BFT.

So far, there is no evidence for the direct interaction of BFT with GPR35. In our previous work, we showed that BFT binds an unknown receptor that triggers γ-secretase activation, resulting in canonical E-cadherin cleavage[18]. Specific binding of BFT requires BFT protease activity[20]. Here, we show that GPR35 senses BFT, and is involved in the downstream actions of BFT, but the binding of BFT to the epithelial cell is maintained in GPR35 knockdown and knockout cells as well as cells treated with the selective GPR35 antagonist ML145, indicating that BFT binding is not mediated by GPR35. In addition, the overexpression of GPR35 in BFT-unresponsive HEK293 cells did not result in increased BFT binding. This points toward a more complex interaction between GPR35 and BFT at the epithelial cell membrane. We hypothesize that ETBF/BFT can trigger the activation of GPR35 signaling through a ligand-dependent and/or -independent mechanism. For example, BFT could trigger the

epithelial cell to release a molecule that serves as a GPR35 ligand to activate GPR35, or GPR35 may be activated under a non-canonical mechanism. Since GPCRs can interact with cytoskeleton and adhesion proteins and BFT can also alter these cellular components[15], this could potentially mediate indirect activation of GPR35. Nevertheless, GPR35 functions downstream of the cellular events following BFT exposure and controls the flow of signaling cascades activated by BFT-receptor interaction. Importantly, activation of GPR35 leads to cellular responses to stress and cell morphogenesis[47], similar to BFT, showing that although binding of BFT is not mediated by GPR35, the signaling networks of BFT and GPR35 at least partly overlap. How GPR35 senses BFT is an important question to further investigate.

The data generated from the ETBF mouse colitis models support our in vitro and ex vivo findings. On day 3 post ETBF colonization, GPR35-knockout mice showed improved clinical responses and survival in two mouse models. The reduced death rate and increased body weights in GPR35-knockout mice compared to wild-type mice suggest that CECs are less responsive to BFT in the absence of GPR35. Altogether, this supports the notion that GPR35 is important in the initial epithelial response to ETBF/BFT in the mouse colon. The only genes that were differentially expressed in GPR35$^{-/-}$ compared to wild-type mice following ETBF infection were Cxcl5, Il-22, and Mt2. These are all important genes in the innate immune response, as CXCL5 stimulates neutrophil recruitment[51], IL-22 is important for host defense and mucosa regeneration[52], and metallothionein MT2 functions as a modulator of the immune response by zinc sequestration, a metal important for the function of immune cells[53]. As downregulation of MT2 also results in a reduced availability of extracellular zinc pools, this may affect the function of cellular metalloproteinases and BFT as well, since BFT is a zinc-dependent metalloprotease[54]. We have previously shown that ETBF induces NF-kB activation and Cxcl5 expression, primarily in the distal colon CECs[9]. These CEC actions of BFT trigger the accumulation of myeloid cells in the distal colon and the onset of the mucosal IL17A response to ETBF, both of which contribute to ETBF procarcinogenesis[9,23,55]. Thus, a full response to ETBF colonization requires the participation of both CECs and the mucosal immune response. Herein, our use of both in vitro and in vivo models allowed us to partially dissect the contributions of GPR35 in CECs and the mucosal immune response to ETBF. Both our in vitro and in vivo GPR35$^{-/-}$ models were consistent in showing biologic delays in the response to BFT (in vitro) or ETBF (in vivo). Additional studies will be needed to more definitively determine the contribution of GPR35 in different cell types to ETBF colonization and signaling. Because our model involves global KO of GPR35 and the literature indicates that GPR35 is important in CEC biology and may play an important role in gut macrophages, additional work is needed to discern the contributions of GPR35 in the CEC and/or immune cell compartments to ETBF pathogenesis[56,57].

Multiple factors could explain why only a reduction in inflammation was observed in GPR35$^{-/-}$ mice compared to wild-type mice at the initial stage of ETBF infection. The gut microbial ecosystem is complex and multiple factors may contribute to the inflammatory response upon ETBF infection. For example, ETBF could be a chaperone of other bacteria in penetrating through the mucus layer to get direct contact with epithelial cells. This was shown recently for the oncogenic microbes ETBF and Escherichia coli producing the genotoxin colibactin, suggesting that ETBF leads the way for DNA-damaging agents and other oncogenic or colitigenic microbes by degrading the mucus layer and stimulating a procarcinogenic Th17 response via colon epithelial cell signaling[9,28]. Thus, infection with ETBF could

drive microenvironmental changes that regulate microbe:CEC interactions via separate GPR35-independent pathways. For example, reduction in E-cadherin expression results in reduced barrier function and changes the microenvironment of colon epithelial cells by creating opportunities for invasion of other gut (opportunistic) microbes that may comediate a colitis response. Overall, our results suggest that intestinal GPR35 guides the mucosal inflammation caused by ETBF and is responsible for sensing BFT in the mucosa. Because GPR35 is also found on some immune cells, it is possible that not only GPR35 on CECs is important for BFT-induced colitis, but GPR35 may also be involved in the interplay between the immune system and CECs following ETBF colonization.

Our results show that GPR35 is an important signaling molecule for mediating the CEC and inflammatory effects to BFT/ETBF. GPR35 polymorphisms on chromosome 2q37 have been found with GWAS in relation to ulcerative colitis[30,58] and Crohn's disease[31]. Importantly, the rs3749171 single-nucleotide polymorphism results in a T108M mutation of the GPR35 protein, which has been shown to not only predispose individuals to ulcerative colitis and PSC but also to cancer risk associated with these disaeses[38]. It is therefore necessary to understand the role of GPR35 polymorphisms in BFT/ETBF signaling. These polymorphisms might render certain patients more susceptible to oncogenic signaling triggered by BFT/ETBF. GPR35 seems likely to be of importance for other (microbial) signaling molecules and may play a more general role in bacterial sensing and outcomes in the colon of patients with IBD.

## Methods

**Human cell culture.** HT29/c1 cells (obtained from Daniel Louvard, Institute Pasteur) were maintained in DMEM medium supplemented with 10% fetal calf serum (Gibco), 10 µg/ml apo-transferrin (Sigma T1147), 50 units/ml penicillin, and 50 µg/ml streptomycin (Gibco). Cells were subcultured every 3–5 days for a maximum of 20 passages.

HEK293T cells (Flp-In T-Rex 293 cells) were stably transfected with FLAG-HumanGPR35-eYFP cDNA (provided by Graeme Milligan). The construct was placed into the pcDNA5/FRT/TO vector and the pOG44 vector (1:9) to inducibly express GPR35 via lentiviral cell infection. Stably transfected cells were maintained in 10% FCS, 10 µg/ml blasticidin, 200 µg/ml hygromycin B, 50 units/ml penicillin, and 50 µg/ml streptomycin (Gibco). GPR35 expression was induced with 100 ng/ml doxycycline for 24–48 h (effective concentration determined by serial dilution).

**Bacterial cell culture.** Bacteroides fragilis strain O86-5443-2-2 containing the BFT-2 toxin was grown anaerobically in Brain Heart Infusion (BHI) broth supplemented with 5 g/L yeast extract, 0.5 g/L cysteine, 5 mg/L hemin, and 1 mg/L vitamin K for 24–48 h for in vitro assays. For mouse inoculation and optimal colonization, 4–5 colonies were selected from BHI agar and grown for 8 h at 37 °C in an anaerobic chamber. Bacterial cells were diluted 1:20 in fresh BHI medium containing 6 µg/ml clindamycin for inoculation. Cells were washed twice with PBS and resuspended in PBS for inoculation at $OD_{600}$ of 0.8–1.0.

**Toxin assay.** The HT29/c1 toxin assay was performed as previously described by Weikel et al.[14] (1992 Infect Immun). In short, HT29/c1 cells were grown until 60–70% confluency on 8-well LabTek slides or 96-well plates. Next, cells were washed with FCS-free DMEM medium twice before the addition of cell-free supernatants of BFT-2-producing strain O86-5443-2-2 or purified BFT-2 toxin[59] in concentrations of 100 ng–200 ng/ml in DMEM without FCS. HT29/c1 cells were exposed to supernatants or purified toxin for 1–3 h. Toxin scoring was performed by two independent reviewers that were blinded for the conditions.

**GPR35 antagonist treatment.** The GPR35 antagonists CID2745687 (100 mM) and ML145 (50 mM) were dissolved in DMSO to create stocks. Concentration series of ML145 in DMEM were tested on HT29/c1 cells from 20 µM in tenfold dilutions to 2 nM with a maximum concentration of DMSO of 0.1%. Cells were pretreated with and without antagonists for 30 min at 37 °C in a humidified incubator. After 30 min, BFT-2 at 200 ng/ml was added to the wells to reach an end concentration of 100 ng/ml BFT-2 per well. As a reference for each antagonist concentration, DMEM without BFT was added. A negative control without toxin and the highest concentration of DMSO (0.1%) was used. Toxin activity was scored 1 and 3 h after incubation with BFT-2 by three independent persons blinded for each condition.

**Table 1 Antibody conditions western blot.**

| Primary antibody | Provider | Dilution | Incubation conditions | Secondary antibody | Dilution | Incubation conditions |
|---|---|---|---|---|---|---|
| Mouse-anti-beta-arrestin 1 | BD (610551) | 1:1000 | O/N 4 °C | Goat-anti-mouse | 1:5000 | 1 h RT |
| Goat anti-beta-arrestin 2 | Abcam (ab31294) | 1:2000 | O/N 4 °C | Donkey-anti-goat | 1:2000 | 1 h RT |
| Rabbit anti-Na$^+$K$^+$ATPase | Thermo Scientific (PA5-17251) | 1:1000 | O/N 4 °C | Goat-anti-rabbit | 1:3000 | 1 h RT |
| Mouse-anti-HDAC2 | Cell signaling (3F3, #5113) | 1:100 | O/N 4 °C | Goat-anti-mouse | 1:3000 | 1 h RT |
| Mouse-anti-beta-actin | Sigma Aldrich (A5441) | 1:10000 | 2 h RT | Goat-anti-mouse | 1:5000 | 1 h RT |
| Rabbit anti-E-cadherin | Gift from James Nelson (Stanford University) | 1:1000 | O/N 4 °C | Goat-anti-rabbit | 1:5000 | 1 h RT |
| Rabbit anti-GFP | Abcam ([E385], ab32146) | 1:1000 | O/N 4 °C | Goat-anti-rabbit | 1:5000 | 1 h RT |
| Mouse-anti-FLAG M2 | Sigma Aldrich (F3165) | 1:1000 | O/N 4 °C | Goat-anti-mouse | 1:5000 | 1 h RT |

**Binding of recombinant BFT to HT29/c1 cells**. HT29/c1 cells or HEK293 cells were seeded in 96-well plates and washed three times with PBS before the addition of HRP–streptavidin (1:500) to block nonspecific biotin sites on epithelial cells. Cells were washed and preincubated with ML145 (20 µM) or vehicle (0.1% DMSO) for 30 min at 37 °C. Next, medium with/without recombinant BFT-1 (rBFT-1[59]) labeled with biotin at concentrations of 300, 100, 30, 10, 3, and 1 ng/ml was added to the cells and incubated for another hour at 37 °C. Biotin-labeled BSA was used as a control to eliminate nonspecific background signal. After incubation, cells were washed three times with PBS, fixed in 4% paraformaldehyde, and blocked with 5% BSA–PBS for 1 h at RT. rBFT-1-biotin was visualized with streptavidin-800ICW (1 h RT in 5% BSA–PBS). Cells were counterstained with DRAQ5 (1:2000) for quantification relative to the cellular density in each well.

**shRNA knockdown (KD) with lentiviruses**. The initial 82 shRNA clones (shown in Supplementary Table 1) were provided by the High Throughput Biology Center (School of Medicine, JHU). The viral particles were packaged in 293T cells using Mission Lentiviral Packaging Mix (SHP001, Sigma Aldrich). The viral titers were determined by HIV-1p24 Antigen ELISA (0801111, ZeptoMetrix Corporation). The additional lentiviral transduction particles were purchased from Sigma Aldrich as GPR35: clone 1 (TRCN0000357167), clone 2 (TRCN0000357166), clone 3 (TRCN0000367796), clone 4 (TRCN0000008889), clone 5 (TRCN0000008891), clone 6 (TRCN0000008890), and beta-arrestin: clone β-arr1 (TRCN0000230149) and β-arr2 (TRCN0000280685). HT29/c1 cells grown on 96-well or 6-well plates were infected for 18–20 h at 37 °C with 5% CO$_2$ with a multiplicity of infection (MOI) between $10^4$ and $10^6$ viral particles. The medium was refreshed with DMEM containing 10% FCS and again incubated for 2–4 days before screening for BFT-induced morphological change. Stably transfected HT29/c1 cells were selected using DMEM medium containing 2 mg/ml puromycin.

**CRISPRcas GPR35-knockout (KO) HT29/c1 cell line**. The GPR35 KO in HT29/C1 cells using CRISPRcas was performed using a lentiviral delivery system with four designed guides (1A, 1B, 2A, and 2B). The guides were generated from genomic DNA of HT29/c1 cells by the following primer set combinations targeting exon#7 of human GPR35 (2q37.3) [guide 1A: GPR1AF (caccgggggccaggtgaggtcgc) and GPR1AR (aaacgcgacctcacctggccccc); guide 1B: GPR1BF (caccgtgggcttctacgcc-tact) and GPR1BR (aaacagtaggcgtagaagcccac); guide 2A: GPR2AF (caccgccagcag-gacgcccaagt) and GPR2AR (aaacacttgggcgtcctgctggc); guide 2B: GPR2BF (caccgtcaacagcctggcgctct) and GPR2BR (aaacagagcgccaggctgttgac)]. Each of the four guide RNAs [1A (gcgacctcacctggcccc), 2A (acttgggcgtcctgctgg), 1B (tgggcttctacgcctact), and 2B (tcaacagcctggcgctct)] were ligated into a lentiviral plasmid backbone lentiCRISPR v2 (Addgene #52961). The clones were sequence-verified and transfected into HEK293 cells along with the packaging plasmids pMD2.6 (Addgene plasmid 12259) and psPAX2 (Addgene plasmid 12260). The viral particles containing GPR35 gRNA were collected from the culture supernatant of HEK293 cells 24–48 h post packaging and used to infect HT29/c1 cells grown on 96-well plates. The positively infected cells were selected using puromycin (2 µg/ml) in the culture medium and were divided into three portions: (1) preserved for later use, (2) diluted and plated on 96-well plate to obtain stable clones, and (3) isolated genomic DNA for initial surveyor test using the primer set: survF3: 5'-GCCCTCCCTGCTAAGAGCTG-3' and survR3: 5'-ATGCCCTGGGAGAGCT GG-3'. Twenty stable clones (five from each of four constructs) were expanded for the second surveyor test. The PCR products amplified by survF/R from 12 clones (two from each of gRNA 1A and 1B and four from each of gRNA 2A and 2B) that were repeatedly positive on the surveyor test were sequenced. Three clones from guides 2A and 2B whose GPR35 gene editing was confirmed through DNA sequencing were used for further biological characterization.

**Protein isolation**. After 1–4 h with or without BFT treatment (100 ng/ml), cells were washed and harvested for total cell extraction in 2% SDS in PBS with protease inhibitor cocktail on ice. For subcellular protein extraction, cells were harvested by trypsinization and centrifuged at 500× $g$ for 10 min. The Thermo Scientific subcellular protein extraction kit (87790) was used for extraction of FI (cytoplasmic), FII (membrane), FIII (nuclear), FIV (chromatin-bound nuclear), and FV (cytoskeleton). Protein concentrations were measured through BCA protein assays (Pierce). HEK293-induced and noninduced cells were lysed in lysis buffer (0.5% Igepal CA630 in 1× PBS with 1× protease inhibitors).

**Western blot**. Bis-tris precast gels (4–12% gradient) were used for gel electrophoresis in MES buffer according to the manufacturer (BioRad). A benchmark prestained protein ladder was used to confirm relative protein size. Gels were run on 140 V for 5–10 min and an additional 35 min at 200 V. Proteins were transferred to a PVDF membrane in blot buffer containing 10% methanol (Invitrogen). PVDF membranes were activated in 100% methanol and proteins were transferred for 1.5 h at 100 V on ice. Membranes were blocked with 5% milk in Tris-buffered saline with tween20 (TBS-T) for all antibodies, except phosphor ERK that was blocked in 5% BSA in TBS-T. Next, primary antibody incubations were performed according to the conditions in Table 1. Subsequently, blots were washed three times for 10 min in TBS-T and incubated with the secondary antibodies as listed in Table 1 (Donkey-anti-goat-HRP (Promega, V0851), goat-anti-rabbit-HRP (Jackson, 111-035-144), and goat-anti-mouse-HRP (Jackson, 115-035-003)). All antibodies were visualized with regular ECL (Invitrogen).

**ELISA (E-cadherin, IL-8)**. E-cadherin ELISA was performed as described by the manufacturer (Quantikine sE-cadherin ELISA kit, R&D Systems). Epithelial supernatants were added in a twofold dilution to the wells in assay buffer for detection of cleaved E-cadherin.

IL-8 ELISA: The capture antibody mouse-anti-human IL-8 (BD Pharmingen 554761) was coated in 0.1 M bicarbonate buffer (pH 9.5) overnight at 4 °C. The plate was washed 3× with 300 µl wash buffer (PBS—0.05% Tween-20). Next, the plate was blocked with assay diluent (PBS (pH 7.0) containing 10% FCS) for 1h at RT. After three washes with wash buffer, standards (1000 pg/ml to 15.6 pg/ml) and cell supernatants (2× diluted) were diluted in assay diluent and applied in duplicate to the ELISA plate and incubated for 2 h at RT. Plates were then washed and detection antibody was added 1:500 (BD Pharmingen 554718) for 1h at RT. After washing, poly-HRP was added at 1:10.000 and incubated for 20 min at RT. The signal was visualized with 100 µl TMB-substrate (Pierce), and the reaction was stopped with 2N H$_2$SO$_4$. Absorbance was read at 450 nm with a reference channel at 655 nm.

**Immunofluorescence and confocal imaging**. β-arrestin: HT29/c1 cells on glass coverslips were exposed to the GPR35 agonist compound 10 at 10 µM or BFT at 100 ng/ml for 30 or 60. After incubation, cells were washed with HBSS and fixed in 4% paraformaldehyde for 20 min. Cells were permeabilized with 1% Triton-X100 in PBS for 5 min. Cells were blocked with 5% BSA in PBS for 1 h at RT. Goat-anti-beta-arrestin-2 (1:100, ab31294) in TBS-T containing 1% BSA or rabbit-anti-beta-arrestin 1 (1:100, ab32099) was incubated overnight at 4 °C. Donkey-anti-goat antibody labeled with Alexa fluor 568 (1:500) (Fisher scientific #A-11057) or goat-anti-rabbit antibody labeled with Alexa 488 (1:500, Fisher scientific, #A-27034) in TBS-T containing 1% BSA was incubated for 1h at RT for visualization. E-cadherin was visualized with 1:200 (E2 antibody) in 5% BSA–PBS-T with secondary goat-anti-rabbit Alexa fluor 568 (Fisher scientific, #A-11011) at 1:500. Cells were counterstained with DAPI (1:10.000) and mounted using prolong Gold anti-fade reagent. Cells incubated with 100 ng/ml BFT-labeled with Alexa 488 were visualized with confocal microscope Olympus FV1000. All other immunofluorescence was imaged on a confocal microscope Zeiss LSM510.

**Real-time PCR**. RNA was isolated from wild-type HT29/c1 cells and stably transfected HT29/c1 cells treated or not with BFT-2 using a Qiagen RNA isolation kit (RNeasy mini kit). For cDNA production, 1 µg of RNA was mixed with 10× RT buffer, 100 mM dNTP Mix, 10× RT random primers, 1 µl of multiscribe reverse transcriptase, and 1 µl of RNAse inhibitor in nuclease-free water (Applied Biosystems). cDNA was generated in a thermal cycler (BioRad) at 25 °C for 10 min,

**Table 2 Primer combination for SYBR green qPCR.**

| Gene | Species | Primer forward (5′→3′) | Primer reverse (5′→3′) | Bp | Detector |
|------|---------|------------------------|------------------------|-----|----------|
| 18S rRNA | Human | TTGACTCAACACGGGAAAC | ACCCACGGAATCGAGAAAGA | 81 | SYBR |
| GPR35 a+b | Human | GATCAAGCTGGGCTTCTACG | CAGGCTGATGCTCATGTACC | 265 | SYBR |
| GPR35b | Human | ACACCGTGGCAGTGAAGAG | CAGGCTGATGCTCATGTACC | 379 | SYBR |
| IL-8[60] | Human | GTTGTAGTATGCCCCTAAGAG | CTCAGGGCAAACCTGAGTCATC | 407 | SYBR |
| GAPDH | Human | TCCTGCACCACCAACTGCTTAG | TGCTTCACCACCTTCTTGATGTC | 341 | SYBR |

**Table 3 Taqman assays for qPCR.**

| Gene | Species | Taqman assay | Bp | Detector |
|------|---------|--------------|-----|----------|
| 18S rRNA | Human | 4310893E | 187 | VIC-TAMRA |
| GPR35 | Human | Hs00271114_s1 | 87 | FAM |
| ARRB2 | Human | Hs01034135_m1 | 58 | FAM |
| ARRB1 | Human | Hs00244527_m1 | 136 | FAM |
| GAPDH | Mouse | Mm99999915_g1 | 109 | FAM |
| Il17a | Mouse | Mm00439619_m1 | 91 | FAM |
| Nos2 | Mouse | Mm00440502_m1 | 66 | FAM |
| Il6 | Mouse | Mm00446190_m1 | 78 | FAM |
| Ifn-γ | Mouse | Mm00801778_m1 | 101 | FAM |
| Tnf-α | Mouse | Mm00443260_g1 | 61 | FAM |
| Il22 | Mouse | Mm01226722_g1 | 65 | FAM |
| Cxcl5 | Mouse | Mm00436451_g1 | 75 | FAM |
| Mt1 | Mouse | Mm00496660_g1 | 88 | FAM |
| Mt2 | Mouse | Mm00809556_s1 | 111 | FAM |

37 °C for 120 min, and 85 °C for 5 min. All real-time assays were amplified on a 7900 HT real-time PCR cycler (Applied Biosystems) relative to the 18S and/or GAPDH reference genes. All primer combinations and Taqman assays are listed in Tables 2 and 3.

**Colon enteroid derivation and culture.** Mouse colons were dissected from wild-type C57Bl6 and GPR35$^{-/-}$ C57Bl6 mice. Two-thirds of the colon, including the distal and middle areas, were minced into small pieces and washed in the CSC buffer. Following an hour incubation in CSC buffer containing 20 mM EDTA with shaking at 4 °C, the tissue pieces were filtered through a 70 μM mesh filter and washed again by centrifugation, then suspended in 50 μl of Matrigel (Corning™ 356255), and seeded in 24-well plates. The growth medium (supplied by The Hopkins Digestive Diseases Basic & Translational Research Core Center) was applied after the Matrigel solidified, and was refreshed every other day. After 7–10 days, the enteroids were expanded and formed branches. When treated with BFT-2 from O86-5443-2-2, the growth medium was removed and replaced with serum-free media. Purified BFT (500 ng/ml) was added to the medium. Morphological changes of enteroids were observed and recorded under the microscope (Nikon E200).

**Mice colonization with ETBF strain O86-5443-2-2 (BFT-2-producing).** The mouse strains used in this study were C57Bl6 purchased from Jackson Laboratory (Bar Harbor, ME, USA) or obtained as littermates of in-house breeding; GPR35 KO mice (Gpr35$^{tm1(KOMP)Vlcg}$) were purchased from the Knockout Mouse Repository (KOMP at UC Davis, CA, USA). The gpr35 gene deletion size is 923 bps from 94,880,070 to 94,879,148 on Chromosome 1.

The ETBF-colonized mouse colitis models have been described previously[23]. In brief, we administered clindamycin (0.1 gr/l) and streptomycin (5 gr/l) or cefoxitin (100 mg/ml) for 3–5 days or 1–2 days, respectively, rested for 24 h (cefoxitin model), followed by oral inoculation with the ETBF strain 086-5443-2-2 (~1 × 10$^7$–10$^8$ bacteria in PBS) or PBS alone (sham control) in mice at 4 weeks of age. Mice were randomly placed in groups, making sure there were no cage effects for mice experiments. Sham mice were housed separately from ETBF-colonized mice. We quantified fecal bacterial colonization as colony-forming units (CFU) per gram stool. At experimental time points, we harvested one piece each of distal and proximal colons in Trizol reagent for RNA analysis. The remainder of the colons were Swiss-rolled, paraffin-embedded, and subsequently sectioned for H&E staining or IHC analysis.

For the ETBF colitis mouse models, GPR35$^{+/+}$, GPR35$^{+/-}$, and GPR35$^{-/-}$ mice were generated by breeding GPR35$^{+/-}$ mice. Body weight loss, indicative of colitis severity, and death rates were monitored for the first 3–7 days of ETBF infection while mice were kept in antibiotic-free clean cages. At the end of the experiment, the remaining mice were sacrificed and colons were collected for RNA or histology analysis.

All mice were kept in specific pathogen-free (SPF) conditions prior to ETBF colonization in the JHU animal facility. The mouse protocols were approved by the Johns Hopkins University Animal Care and Use Committee in accordance with the Association for Assessment and Accreditation of Laboratory Animal Care International.

**Histology, inflammation, and hyperplasia scoring.** Paraffin-embedded sections of Swiss-rolled mouse colons were stained by H&E (hematoxylin and eosin) and scored by a pathologist that was blinded for the conditions for inflammation and hyperplasia. Inflammation was graded from 0 to 4 as 0 (normal mucosa), 1 (mild increase in inflammatory cells, no mucosal changes), 2 (moderate increase in inflammatory cells, mild scattered proliferation focal loss of crypt architecture), 3 (severe increase in inflammatory cells, diffuse or nearly diffuse proliferation, and focally extensive loss of crypt architecture), and 4 (complete or nearly complete mucosal destruction). The hyperplasia was graded from 0 to 3 as 0 (normal), 1 (patchy distribution of mildly deepened crypts and slightly thicker mucosa), 2 (moderate crypt length with hyperchromatic CEC and goblet cell loss), 3 (severe, more than twofold, increase in crypt length with arborized crypts, and high mitotic index such as branched crypts). Anoikis (crypt epithelial cell shedding) was noted.

**Statistics and reproducibility.** For bar graphs of in vitro experiments concerting expression levels, toxin scores, and IL-8-protein changes, independent one-sample t tests were performed (two-sided). Bar graphs are plotted with mean and standard deviations and individual data points of independent experiments are presented. Concentration series with ML145 with readouts for toxin scores and BFT binding at different time points were evaluated with 2-way ANOVA with Dunnett's multiple comparison test. A p-value below 0.05 was considered significant. Each experiment was repeated at least once independently. The majority of the experiments were repeated three times. No data were excluded from the in vitro analysis. Experimental wells for in vitro cell experiments were randomly placed in groups, making sure there was no batch effect of experimental plates. The number of replicates is noted in each graph and figure legends.

Survival data were compared with the log-rank (Mantel) cox test. Body weight changes between sham and the experimental groups in GPR35 wild-type, GPR35$^{+/-}$, and GPR35$^{-/-}$ mice were compared with two-way ANOVA and Bonferroni post tests to compare individual columns. Box–Whisker plots of the median, range, and first and third quartile are plotted. The gene expression data between wild-type and GPR35$^{-/-}$ mice were compared with Mann–Whitney U test, and data were plotted as median with interquartile range. A P value below 0.05 was considered significant. Mice experiments were performed at least three times for each time point, resulting in n = 10+ per group from three independent experiments. One sham mouse was excluded from analysis because it was cross-infected with an unknown pathogen. The number of mice per group is defined in the figure legends.

**Software and data analysis.** Bar graphs and plots were generated in GraphPad Prism (version 6, GraphPad Software Incorporated). Point mutation matrices (PAM250, neighbor-joining) and protein alignments using Clustal W were created in JALview version 2.11.1.0 (www.jalview.org). Western blot density measurements were performed in Photoshop 2020 (Adobe systems incorporated). Fluorescence images were processed in Fiji 1.51n (Wayne Rasband, National Insitute of Health).

**Reporting summary.** Further information on research design is available in the Nature Research Reporting Summary linked to this article.

## Data availability

All data are available within the paper and Supplementary information. The source data for the main figures are available in Supplementary Data 2. Source data are stored at the secure servers of Radboudumc and Johns Hopkins Medical Institutions. Material requests and correspondence about data can be addressed to Annemarie.Boleij@radboudumc.nl.

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

## Acknowledgements

We thank Saraspadee Mootien from L2-diagnostics LLC for providing us purified recombinant BFT protein. We thank Joanna Melia for her thoughtful discussion. This work was supported by NIH NCI grant GPR35: Role in inflammation and Oncogenesis (R01 CA179440), The Hopkins Digestive Diseases Basic & Translational Research Core Center (P30DK089502), the Bloomberg-Kimmel Institute for Cancer Immunotherapy (90068026), and the Biotechnology and Biosciences Research Council UK (grant number BB/P000649/1). AB was supported by the Rubicon (825.11.031) and VENI-programme (016.166.089) of the Dutch research council (NWO).

## Author contributions

S.W., A.B., and C.L.S. were involved in the design of the study. A.B., P.F., J.A., X.W., S.W., and F.H. performed experiments. A.B., S.W., and C.L.S. drafted the paper. W.D. and B.P. helped creating CRISPRcas KO cell lines. A.M., L.J., and G.M. created GPR35 constructs for overexpression of GPR35 in HEK293T cells and provided input and help with GPR35 inhibitors and signaling response. D.H., S.B., and R.A.A. scored histology of mice colonic tissue, all authors gave input and agreed with the final paper.

## Competing interests

C.L.S. receives unrelated research funding from Bristol Myers Squibb and Janssen. The remaining authors declare no competing interest.
