## [Peer Review File · Communications Biology]

Reviewers' comments:

Reviewer #1 (Remarks to the Author):

Boleij et al have investigated the role of GPR35 receptor in colonic epithelial responses to enterotoxigenic *B. fragilis* (ETBF). GWAS data has suggested for a role for GPR35 in IBD pathogenesis. The *B. fragilis* toxin (BFT) plays a role in IBD and colorectal cancer. In a series of in vitro immortalized cell line studies, colonic enteroids and antibiotic treated colitis models in vivo, a robust evidence for at least a partial role of GPR35 in epithelial responses to BFT has been provided. The findings are novel and will be of interest to a range of disciplines including IBD, CRC and infectious diseases.

1. Authors should provide little more introduction on the role of BFT in IBD and colorectal cancer pathogenesis. This is essential to set the stage for examining BFT-GPR35 interactions. Is there known literature on toxin abundance and are there strain variations within *B. fragilis* as it relates to these chronic GI sequelae? Are there non toxin mediated virulence effects as well? The context to study BFT is somewhat lacking.
2. The initial data on 82 candidate proteins and cell morphology assays that led to selection of GPR35 and 3 other targets should be detailed more in the main text.
3. Fig 2- Specificity of ML145 should be expanded upon. How specific are E-cadherin expression and IL-8 production for GPR35 signaling? 1 and 3 hr are early effects as pointed out in paper as well. Did the authors look at later time points (12 or 24 hrs)? Also in enteroids (Fig 3)
4. In vivo studies- to strengthen the in vitro findings, did authors look at E-cadherin and IL-8 expression in GPR35+/+ and GPR35-/- ETBF mice? Was there septicemia/increased bacterial translocation in GPR35+/+ mice? Considering global ko, can authors conclusively say it's the epithelial GPR35 vs effects mediated by immune cell (NK) or PBMC?
5. Antibiotic induced microbial dysbiosis can alter tryptophan metabolism and hence kynurenic acid levels which is a ligand for GPR35. Could this be a limitation of antibiotics treated model? Another possibility is to explore gnotobiotic mice. Authors should report colonic epithelial GPR35 expression in these mice
6. The study will benefit from translational supportive data looking at colonic BFT levels and GPR35 expression in patient biopsies with IBD or CRC.

Reviewer #4 (Remarks to the Author):

The author had clearly described the role of GPR35 by using GPR35-/- and GPR35+/+ mice. The I122, Cxcl5 and Mt2 differences align with the findings. The role of Beta Arrestins 1 and 2 was depicted in the Supplemental data, however in the result section (figure 3) the Beta arrestin 2 data is provided, its acceptable however the rationale should be mentioned in the text. (published article- GPCR signaling via β -arrestin-dependent mechanisms - can be added)

The binding of the BFT the results obtained were negative however an alternative suggestive pathways are not discussed. The GPR35 knockout in HT29/c1 cells and mouse organoids resulted in delaying the BFT response, the link to the experimental data to this experiment in the mice is not highlighted nor it sufficiently discussed. Micro-environmental changes due to E-cadherin down regulation is an interesting part, few lines on this could have been added in the discussion.

Line 251 - Supplemental figure 2d is missing.

The methodology is very clear and the experiments are well planned, using lentivirals for expression and shRNA knockdown, use of antagonists, CRISPR-CAS GPR35 knockout are highly appreciated with matched controls.

A graphical abstract can capture the content of the article for readers at a single glance.

Reviewer #5 (Remarks to the Author):

In the manuscript "G-protein coupled receptor 35 (GPR35) regulates the colonic epithelial cell response to enterotoxigenic *Bacteroides fragilis*" by Boleij et al., the authors performed an exhaustive and comprehensive study to determine if GPR35 is relevant in ETBF-mediated cell changes (i.e., E-cadherin cleavage, cell rounding, IL-8 secretion) and in ETBF-induced colitis. A diverse array of experimental approaches was used to show that inhibition of GPR35, by knockdown assays and inhibitor assay, decreased effects of BFT in vitro. Confirmation of these results was conducted using GPR35 knockout mice which also showed physiologic relevance. The data presented herein supports the authors' conclusions and the data appears solid.

Major comments

1. The GRP35 KD cells show diminished cell morphology change, E-cadherin cleavage, and IL-8 secretion after BFT treatment. It is my understanding from reading the author's previously published papers that extracellular cleavage of E-cadherin is quite rapid - occurring within minutes after BFT treatment. The in vitro data shown by the authors are mostly 3-4hrs post-treatment. Did the authors examine E-cadherin cleavage in a shorter time frame?

Minor comments

1. Page 5, line 127. "Hela" should be "HeLa"
2. Page 19, line 463. "chaperon" should be "chaperone"
3. I was unable to locate supplemental table 1. Although it is not critical for evaluating the manuscript, I wonder if it was lost during the submission.
4. Page 14, line 342: "noet" should be "not"

Point by point response to the Reviewers' comments:

Reviewer #1:

Boleij et al have investigated the role of GPR35 receptor in colonic epithelial responses to enterotoxigenic *B. fragilis* (ETBF). GWAS data has suggested for a role for GPR35 in IBD pathogenesis. The *B. fragilis* toxin (BFT) plays a role in IBD and colorectal cancer. In a series of in vitro immortalized cell line studies, colonic enteroids and antibiotic treated colitis models in vivo, a robust evidence for at least a partial the role of GPR35 in epithelial responses to BFT has been provided. The findings are novel and will be of interest to a range of disciplines including IBD, CRC and infectious diseases.

Response: We thank the reviewer for the positive feedback on our manuscript.

1. Authors should provide little more introduction on the role of BFT in IBD and colorectal ca pathogenesis. This is essential to set the stage for examining BFT-GPR35 interactions. Is there known literature on toxin abundance and are there strain variations within *B. fragilis* as it relates to these chronic GI sequelae? Are there non toxin mediated virulence effects as well? The context to study BFT is somewhat lacking.

*Response: In the introduction we have tried to add more detail on what is known on BFT in IBD, CRC, colonization and pathogenesis. To measure toxin abundance, ideally one would need to measure protein levels; however, BFT is very active at very low concentrations (nM, even fM) and because of the complexity of fecal samples and a preference for the mucosal niche, no reports have been published so far on toxin abundance in humans. We do know, however, that toxin production from different bft isotypes and bft strains varies in our in vivo models (unpublished). We have also measured IgG-responses to BFT in patient sera and see increased detection in colonized patients. Hence, in vivo production of BFT is likely happening and in such a way that it produces an immunological reaction (unpublished observations). So far, to our knowledge, BFT is the only known virulence factor of *B. fragilis* related to chronic inflammation. We have added this detail to the text, including relevant information on toxin expression.*

To introduce important information on BFT earlier in the paper, we have moved the specific information on BFT towards the front of the introduction and added the following information to better describe the context of BFT:

“Several studies have identified an association between enterotoxigenic *Bacteroides fragilis* (ETBF) that secretes the *Bacteroides fragilis* toxin (BFT) and human IBD as well as colorectal cancer (CRC).¹⁻³ ETBF is associated with acute diarrhea in children and adults and asymptomatic colonization is likely frequent, at least, in some populations (~ 0-55%, depending on the study).⁴⁻⁶ It is likely that multiple *B. fragilis* strains can coexist in the GI-tract and that competition between these strains is determined by their adherence and other virulence factors. The pathogenesis of ETBF is dependent on secretion of BFT, a zinc-binding metalloprotease.⁷⁻⁹ *B. fragilis* with a deleted bft-gene is unable to cause inflammation in mouse colitis models.¹⁰ Three different BFT isoforms have been identified and named as bft-1, bft-2, and bft-3¹¹, whose encoded amino acid sequences are >93% identical.^{11,12} Thus, BFT production and activity might depend on the *B. fragilis* strain and secreted BFT isotype in vivo. While bft expression can be inhibited by glucose and fermentable carbohydrates present in the GI-tract, oxygen and the cysteine protease fragipain (Fpn) present in high concentrations in the mucus layer upregulate bft expression potentially enabling successful colonization in the gut mucosa.^{13,14}”

AND

“Recently, several reports have suggested that the G protein-coupled receptor (GPR35) may be related to gastrointestinal inflammation and colitis.”

2. The initial data on 82 candidate proteins and cell morphology assays that led to selection of GPR35 and 3 other targets should be detailed more in the main text.

Response: To better explain the selection of GPR35 we have included a section on the microarray subtraction assay and following shRNA KD experiments in the Results section:

A microarray data subtraction comparison among the BFT-responsive CRC epithelial cell lines HT29/c1 (in this study), HT29 (GSM396550), T84 (in this study), Caco-2 (GSM24832) and SW480 (in this study) versus the BFT-non-responsive kidney cell line HEK293 (in this study) and HeLa cells (GSM410912) was conducted and identified 82 epithelial membrane-related proteins as potential receptor candidates (Supplementary table 1). We used lentiviral shRNA knock-down (KD) screening of these 82 proteins to identify candidate proteins that diminished or blocked BFT-induced morphological changes in HT29/c1 cells (details on shRNA infection see below). After 48 hours of shRNA KD, morphological change of the 82 shRNA KDs with and without BFT treatment (5nM) were monitored for 24 hours. Based on this initial screening 3 clones emerged as they showed reduced morphological change; [c1-8 (GPR35; TRCN0000008887), c6-9 (Claudin-4; TRCN0000116627), and c3-7 (RAB20; TRCN0000048093)] (Supplementary table 2). These three clones and the membrane receptor CD44 (TRCN0000057563) as a control were selected to generate stable cell lines; morphology assay and E-cadherin western blot were performed with and without BFT for 1-3 hours to confirm the loss of BFT activity in the stable cell line clones.”

3. Fig 2- Specificity of ML145 should be expanded upon. How specific are E-cadherin expression and IL-8 production for GPR35 signaling? 1 and 3 hr are early effects as pointed out in paper as well. Did the authors look at later time points (12 or 24 hrs)? Also in enteroids (Fig 3)

Response: ML145 has been shown to be an effective antagonist with a high affinity for human GPR35 and acting as a competitive antagonist for a number of GPR35 agonists (Heynen-Genel et al. 2010). To highlight this selectivity, we have added the following sentence to the manuscript:

“ML145 has a high affinity for human GPR35 and, for example, has a >1000-fold higher affinity towards GPR35 than GPR55; ML145 is the most selective known antagonist for GPR35.⁴³”

E-cadherin cleavage and IL-8 production are not specific for GPR35 signaling per se, but are specific for BFT. We reasoned that if GPR35 plays a role in early signaling of BFT from the cell surface GPR35KD would interfere with E-cadherin cleavage and IL-8 secretion. A recent publication into agonists of GPR35 (pamoic acid and YE210) to identify the GPR35 network of signaling events suggests that activation of GPR35 is related to morphological processes, such as cytoskeletal remodeling and has a function in inflammation, both of which are also affected by BFT. We have added this information to the discussion:

“Furthermore, a recent study investigating the signalling network of GPR35 in HT29 cells shows that stimulation of GPR35 with agonists pamoic acid and YE210 is related to changes in morphological processes, such as cytoskeletal remodelling and has a function in inflammation⁵⁰, both of which are also affected by BFT.”

Indeed 1 and 3 hours after BFT-treatment are early effects. The morphological change of treated HT29/C1 cells is clearly visible at 1 hour and progresses over time and is usually maximal by 3 hours and still visible at 24h (Weikel et al 1992). BFT-induced cell morphology changes are dependent on the concentration of BFT; the higher the concentration the earlier a maximum cellular effect is reached. After ~24-48h of BFT cell exposure, while cell morphologic changes induced by BFT recover, there is increased colon epithelial cell c-myc expression and proliferation (Wu et al. 2003). Depending on the BFT concentration and duration of BFT storage at -80°C E-cadherin cleavage can be detected within minutes or somewhat later. Our starting point in most experiments is usually ~15-30 minutes, but significant cleavage can generally be observed at 1h and is usually complete at 3h after cell treatment with BFT. These are the reasons why E-cadherin cleavage and morphological changes were studied at these time points in this study. IL-8 mRNA expression also appears rapidly and starts to cease after 6h whereas IL-8 secretion into the cell culture media starts around 3h-4h and increases to 18h (Wu 2004). Also here, early changes are captured. We performed a few experiments at 24 hours but did not observe differences in IL-8 secretion. Using our murine models, we continue to study the role of GPR35 in the chronic phase of ETBF infection.

4. In vivo studies- to strengthen the in vitro findings, did authors look at E-cadherin and IL-8 expression in GPR35+/+ and GPR35-/- ETBF mice? Was there septicemia/increased bacterial translocation in GPR35+/+ mice? Considering global ko, can authors conclusively say it's the epithelial GPR35 vs effects mediated by immune cell (NK) or PBMC?

Response: In mice we focused on the colon inflammatory response, as this is the primary result of ETBF colonization in the colitis model. In our mRNA expression analysis we homogenized small pieces of the distal colon that included all cell types. Hence our reported expression changes are not specific for epithelial cells. As the reviewer requested, we have stained 3 ETBF-colonized GPR35^{+/+} and GPR35^{-/-} mouse distal colon tissues with an E-cadherin antibody and compared these to sham control mice. We observed locally reduced E-cadherin staining in colon epithelial cells in all ETBF-colonized mice in both groups. Because the reduced E-cadherin staining is not uniformly throughout the whole colon, it is hard to conclude whether there is a different degree of reduction between the two genotypes. The data have been added to Supplementary figure 4. The chemokine IL-8 is not expressed by mice, and the equivalent chemokine (MIP) of mice is not expressed solely by the epithelial cell, hence this was not assessed.

To address the point on septicemia: in one experiment, we performed bacterial cultures from blood of C57BL/6 mice at 2 and 3 days post colonization from ETBF. We obtained positive cultures in only 2/11 (18%) ETBF mice. Therefore, we did not pursue this evaluation in this study.

Because of the global KO of GPR35, we can indeed not definitively define the potential contributions of epithelial vs immune cell GPR35 in our model. From the literature, it is evident that GPR35 is expressed in the proximal and distal colon, and on macrophages, and less expressed in antibiotic-treated animals or germ-free mice (Kaya 2020 et al.). To begin to evaluate GPR35 on epithelial cells, in one experiment, we performed a bone marrow chimera model by transplant of wild type C57BL/6 BM to GPR35 KO mice and vice versa after lethal radiation. The preliminary flow cytometry data of this chimera experiment suggest that GPR35 plays a role in both the epithelial and immune compartment of the mice 7 days post-colonization (see figure below). The method we used here for the flow cytometry is described in Housseau et al. *Cancer Res* 2016. We plan to continue to study the effects of both compartments in the future but consider this topic to be quite complex and beyond the scope of this initial manuscript. To raise the reviewer's excellent point with the reader, we have included in the discussion the potential role of both GPR35 on immune cells and epithelial cells.

- A. Higher percentages of IL17- producing lymphocytes (IL17tat) and Th17 cells were detected in GPR35^{-/-} mice that received GPR35^{+/+} BM (group 4) compared to the GPR35^{-/-} mice that received GPR35^{-/-} bone marrow (group 3) at day 7 post-ETBF colonization by flow cytometry. The GPR35^{+/+} mice (wt) that received GPR35^{-/-} bone marrow (group 5) showed elevated IFN γ in lymphocytes (IFN γ tat) and in cytotoxic T cells (TC1) compared to the same mice that received GPR35^{+/+} bone marrow (group 2). These data point out a potential role of GPR35 on epithelial and immune cells in regulating the mucosal Th1 and Th17 response.
- B. Higher percentage of Mac2 (MHCII low) and monocytic myeloid cells were detected in the colon mucosa of GPR35^{-/-} mice that received GPR35^{+/+} bone marrow (group 4, i.e., GPR35^{-/-} KO colon epithelial cells (CECs) and wild-type (wt) BM) compared to GPR35^{+/+} control mice (group 2, wt CEC and wt BM). These two subsets of cells are also highly detected in the GPR35^{+/+} mice that received GPR35^{-/-} bone marrow (group 5, wt CEC and KO BM). The data suggest that the reduced immune cell recruitment observed in GPR35^{-/-} mice (group 3, GPR35^{-/-} KO CEC and KO BM) results from the combined effect of GPR35 deficiency on both immune and epithelial cells.

Added to the manuscript Discussion: "Because our model involves global KO of GPR35 and the literature indicates that GPR35 is important in CEC biology and may play an important role in gut macrophages, additional work is needed to discern the contributions of GPR35 in the CEC and/or immune cell compartments to ETBF pathogenesis (Kaya 2020 et al. and Tsukahara et al. 2018)."

5. Antibiotic induced microbial dysbiosis can alter tryptophan metabolism and hence kynurenic acid levels which is a ligand for GPR35. Could this be a limitation of antibiotics treated model? Another possibility is to explore gnotobiotic mice. Authors should report colonic epithelial GPR35 expression in these mice

Response: As the reviewer knows, antibiotic-treated mouse colitis models are very commonly used models. In our model, the antibiotic treatment is short and not continued during our experiments and our sham control group is treated similarly with antibiotics. Whether and how much our antibiotic treatment affects kynurenic acid levels in the mouse gut is unknown. As the reviewer suggests, kynurenic acid levels might be changed due to the antibiotic treatment, however, because all the mice are treated equally, we do not expect the observed effects to be the result of changed kynurenic acid levels. In a preliminary experiment, GPR35 expression is detectable in the colon of germ-free mice. However, germ-free mouse models, which we agree could address this point, are not feasible with ETBF as the bacterium is lethal to germ-free mice a few days after colonization (Rhee et al. 2009).

6. The study will benefit from translational supportive data looking at colonic BFT levels and GPR35 expression in patient biopsies with IBD or CRC.

Response: We agree with the reviewer. We continue to work on this topic with our patient cohorts from the USA and the Netherlands. We expect to report our results in a follow-up manuscript.

Reviewer #4 (Remarks to the Author):

The author had clearly described the role of GPR35 by using GPR35^{-/-} and GPR35^{+/+} mice. The Il22, Cxcl5 and Mt2 differences align with the findings. The methodology is very clear and the experiments are well planned, using lentivirals for expression and shRNA knockdown, use of antagonists, CRISPR-CAS GPR35 knockout are highly appreciated with matched controls.

1. The role of Beta Arrestins 1 and 2 was depicted in the Supplemental data, however in the result section (figure 3) the Beta arrestin 2 data is provided, its acceptable however the rationale should be mentioned in the text. (published article- GPCR signaling via β -arrestin-dependent mechanisms - can be added)

Response: Thank you for this input. To address the reviewer's point, we included statements to address the involvement of beta-arrestin2 in GPR35 signaling (i.e., beta-arrestin2 is more correlated with GPR35 signaling based on the literature) and also added a reference on GPCR-signaling via beta-arrestin-dependent mechanisms.

“G-protein-coupled receptor (GPCR) signalling involves the recruitment of beta-arrestins (β -arr) to desensitize and internalize GPCRs.⁴⁴ Recent research indicates that β -arrestins can also function as signal transducers for GPCRs and result in activation of MAPK and ERK pathways via interaction with Src family kinases.⁴⁵ For GPR35, β -arr2 is considered the most significantly involved in response to agonists.⁴⁶”

2. The binding of the BFT the results obtained were negative however alternative suggestive pathways are not discussed.

Response: We have previously demonstrated that BFT binds to a specific, but unknown, HT29/C1 cell membrane receptor and that specific BFT binding requires BFT protease activity (Wu et al. 2006 & 2007). Here we show that GPR35 senses BFT and is involved in the down-stream actions of BFT, but we also show that GPR35 is not the BFT receptor, confirmed by the BFT binding experiments included in the paper. Based on recent GPR35 literature (Heide Habei Hu 2017), it is likely that GPR35 stimulation is related to the response to stress, the immune system and/or cell morphogenesis. Importantly, activation of GPR35 leads to cytoskeletal remodeling, similar to BFT, showing that

although binding of BFT might not be through GPR35, the signaling networks of BFT and GPR35 at least partly overlap.

We have included this information in the discussion of the manuscript:

“In our previous work, we showed that BFT binds an unknown receptor that triggers γ -secretase activation resulting in E-cadherin cleavage.¹⁹ Specific binding of BFT requires BFT protease activity.²¹ Here we show that GPR35 senses BFT, and is involved in the down-stream actions of BFT, but we also show that binding of BFT to the epithelial cell is not mediated by GPR35, but is maintained in GPR35 knock-down and knock-out cells as well as cells treated with the selective GPR35-antagonist ML145.”

AND

“GPR35 stimulation by agonists is related to cellular response to stress, the immune system and cell morphogenesis.⁴⁷ Importantly, activation of GPR35 leads to cytoskeletal remodelling, similar to BFT, showing that although binding of BFT is not mediated by GPR35, the signalling networks of BFT and GPR35 at least partly overlap. How GPR35 senses BFT is an important question to investigate. We hypothesize that ETBF/BFT can trigger the activation of GPR35 signalling through a ligand-dependent and/or -independent manner.”

3. The GPR35 knockout in HT29/c1 cells and mouse organoids resulted in delaying the BFT response, the link to the experimental data to this experiment in the mice is not highlighted nor it sufficiently discussed.

*Response: We thank the reviewer for pointing this out. As the reviewer appreciates, the *in vitro* data and mouse organoids are a simplified model to study direct interactions of BFT with the epithelial cell and colonic crypts of the gut. It lacks the complexity of a mouse, where also the immune compartment plays an important role. From our previous work in our ETBF *Apc^{min/+}* mice model, we clearly showed the importance of the colon epithelial cell in the colon IL17 response (CHM 2018) where BFT induces a procarcinogenic signaling relay from the CEC where *Cxcl1*, *Cxcl2* and *Cxcl5* expression selectively in distal colon CECs contributes ultimately to a mucosal Th17 response. Thus, the two models are complementary (see comment to reviewer 1 above). We therefore added this information to the discussion of the manuscript.*

*“We have previously shown that ETBF induces NF κ B activation and *Cxcl5* expression, primarily in the distal colon CECs,⁹ These CEC actions of BFT trigger the accumulation of myeloid cells in the distal colon and the onset of the mucosal IL17A response to ETBF, both of which contribute to ETBF pro-carcinogenesis.^{9,23} Thus, a full response to ETBF colonization requires the participation of both CECs and the mucosal immune response. Herein, our use of both *in vitro* and *in vivo* models allowed us to partially dissect the contributions of GPR35 in CECs and the mucosal immune response to ETBF. Both our *in vitro* and *in vivo* GPR35^{-/-} models were consistent in showing biologic delays in the response to BFT (*in vitro*) or ETBF (*in vivo*). Additional studies will be needed to more definitively determine the contribution of GPR35 in differing cell types to ETBF colonization and signalling. Because our model involves global KO of GPR35 and the literature indicates that GPR35 is important in CEC biology and may play an important role in gut macrophages, additional work is needed to discern the contributions of GPR35 in the CEC and/or immune cell compartments to ETBF pathogenesis.”*

4. Micro-environmental changes due to E-cadherin down regulation is an interesting part, few lines on this could have been added in the discussion.

Response: We thank the reviewer for the suggestion. It is true that besides mucus degradation, E-cadherin loss will also result in different host-microbe interactions and give opportunities for other bacteria to invade. To address this point we have added the following sentence to the discussion:

“For example, reduction in E-cadherin expression results in reduced barrier function and changes the micro-environment of colon epithelial cells by creating opportunities for invasion of other gut (opportunistic) microbes that may co-mediate a colitis response.”

5. Line 251 - Supplemental figure 2d is missing.

Response: We thank the reviewer for the comment. We accidentally mislabeled the supplemental figure 2 b and c in the text of the manuscript and have now corrected this.

6. A graphical abstract can capture the content of the article for readers at a single glance.

Response: We thank the reviewer for the suggestion. A graphical abstract was created and added to the manuscript.

Reviewer #5 (Remarks to the Author):

In the manuscript “G-protein coupled receptor 35 (GPR35) regulates the colonic epithelial cell response to enterotoxigenic *Bacteroides fragilis*” by Boleij et al., the authors performed an exhaustive and comprehensive study to determine if GPR35 is relevant in ETBF-mediated cell changes (i.e., E-cadherin cleavage, cell rounding, IL-8 secretion) and in ETBF-induced colitis. A diverse array of experimental approaches was used to show that inhibition of GPR35, by knockdown assays and inhibitor assay, decreased effects of BFT in vitro. Confirmation of these results was conducted using GPR35 knockout mice which also showed physiologic relevance. The data presented herein supports the authors’ conclusions and the data appears solid.

Major comments

1. The GPR35 KD cells show diminished cell morphology change, E-cadherin cleavage, and IL-8 secretion after BFT treatment. It is my understanding from reading the author’s previously published papers that extracellular cleavage of E-cadherin is quite rapid - occurring within minutes after BFT treatment. The in vitro data shown by the authors are mostly 3-4hrs post-treatment. Did the authors examine E-cadherin cleavage in a shorter time frame?

Response: Depending on the BFT concentration and duration of storage, E-cadherin cleavage can be detected within minutes or somewhat later. Our starting point in most experiments is usually ~15-30 minutes, but significant cleavage can generally be observed at 1h and is usually complete at 3h post-infection. These are the reasons why E-cadherin cleavage and morphological changes were studied at 1 and 3h time points.

Minor comments

1. Page 5, line 127. “Hela” should be “HeLa”
2. Page 19, line 463. “chaperon” should be “chaperone”
3. I was unable to locate supplemental table 1. Although it is not critical for evaluating the manuscript, I wonder if it was lost during the submission.
4. Page 14, line 342: “noet” should be “not”

Response: We have corrected these minor mistakes and will make sure the supplemental table is uploaded together with the manuscript.

REVIEWERS' COMMENTS:

Reviewer #1 (Remarks to the Author):

The authors have provided a detailed and satisfactory response to all of my points. This included additional experiments staining 3 ETBF-colonized GPR35+/+ and GPR35-/- and control mice distal colon tissues with an E-cadherin antibody as well as looking for septicemia. Unfortunately the results were not as impressive. However, the effort made is noted and will provide readers better understanding of the system. The contribution of immune and epithelial cell is also addressed including a preliminary experiment in bone marrow chimera model. Appropriate additions have been made to the results highlighting identification of the 3 targets including GPR35 from the 82 candidate proteins. Discussion has been edited to highlight limitations of the global ko. Perhaps addition of a statement near conclusions that "future studies will need to investigate colonic BFT levels and GPR35 expression in patient biopsies with IBD or CRC" might be helpful for readers to reflect on the clinical relevance of the work. I have no further comments for the authors. I believe this will be impactful manuscript for the field and open further line of investigations as well as development of possible preventative/therapeutic targets towards GPR35.

Reviewer #3 (Remarks to the Author):

The authors have sufficiently addressed reviewers' comments. I appreciated inclusion of the graphical abstract. I recommend this article for publication.

Reviewer #4 (Remarks to the Author):

BFT signals through GPR35, the crux of the paper its novel, and the author had identified the target initiated with microarray experiments, validated in cell lines using E cadherin and IL8 with/without shRNA, Using GPR35 agonist, the involvement of beta-arrestin 1 and 2 is interesting to note, BTF delayed response in KO using CRISPR-Cas, animal model with enteroids to prove the concept as in the title. The work is well planned and executed well.

The results start with the supplementary data, not sure the reader will be at ease to do it. A brief method of chip details and time points of microarray can be added in the text.

The revised manuscript has answered all the queries similar queries I had in mind well, it is highly appreciated.

GPR35 do not bind with BFT, more discussion will be expected from the readers, some additional information can add value.

Insilico experiments were mentioned, software used is missing and basic details can be given.

The graphical abstract can be visually improved if a possible bit cluttered. (not necessary)

Minor corrections

The Acronym in the introduction

GWAS - genome-wide association study, not Whole-genome association studies

PSC- Primary sclerosing cholangitis, not sclerosing cholangitis

Pg 13 CRISPCas9 - was used, other places its CRISPCas

Figure 6b also in graphical abstract the colors used are very close and has no contrast - can be improved.

Fig 4-b the contrast can be improved if possible.

Reviewer #5 (Remarks to the Author):

This reviewer is satisfied with the detailed reponse provided by the authors. I have no further comments.

Reviewer #4 (Remarks to the Author):

BFT signals through GPR35, the crux of the paper its novel, and the author had identified the target initiated with microarray experiments, validated in cell lines using E cadherin and IL8 with/without shRNA, Using GPR35 agonist, the involvement of beta-arrestin 1 and 2 is interesting to note, BTF delayed response in KO using CRISPR-Cas, animal model with enteroids to prove the concept as in the title. The work is well planned and executed well.

The results start with the supplementary data, not sure the reader will be at ease to do it. A brief method of chip details and time points of microarray can be added in the text.

Response: we have added a little more detail to the results section of the manuscript: "Total RNA was extracted using Direct-zol RNA kit (ZYMO research) from cultured cells, and the microarray was performed using Human transcriptome array 2.0 (Affymetrix) by the JHMI Deep Sequencing & Microarray Core facility." , and the supplementary data 1 and Supplemental table 1 in the supplementary Information have been updated with number of experiments performed in transient and stable cell lines for these shRNA screens.

The revised manuscript has answered all the queries similar queries I had in mind well, it is highly appreciated.

GPR35 do not bind with BFT, more discussion will be expected from the readers, some additional information can add value.

Response: We have added a little more discussion on how BFT would interact with GPR35 in the discussion of the manuscript: "We hypothesize that ETBF/BFT can trigger the activation of GPR35 signalling through a ligand-dependent and/or -independent mechanism. For example, BFT could trigger the epithelial cell to release a molecule that serves as a GPR35-ligand to activate GPR35, or GPR35 may be activated under a non-canonical mechanism. Since GPCRs can interact with cytoskeleton and adhesion proteins and BFT can also alter these cellular components¹⁵, this could potentially mediate indirect activation of GPR35. Nevertheless, GPR35 functions downstream of the cellular events following BFT exposure and controls the flow of signaling cascades activated by BFT-receptor interaction. Importantly, activation of GPR35 leads to cellular responses to stress and cell morphogenesis⁴⁷, similar to BFT, showing that although binding of BFT is not mediated by GPR35, the signalling networks of BFT and GPR35 at least partly overlap. How GPR35 senses BFT is an important question to further investigate. "

Insilico experiments were mentioned, software used is missing and basic details can be given.

Response: the software was mentioned in the reporting summary, but indeed not in the main manuscript. We have added a section on software used in the methods section including the program and used tools to generate the point mutation matrix and multiple protein alignment.

Software and data analysis

Bar graphs and plots were generated in GraphPad Prism (version 6, GraphPad Software Incorporated). Point mutation matrixes (PAM250, neighbor joining) and protein alignments using Clustal W were created in JALview version 2.11.1.0 (www.jalview.org). Western blot density

measurements were performed in Photoshop 2020 (Adobe systems incorporated). Fluorescence images were processed in Fiji 1.51n (Wayne Rusband, National Institute of Health)."

The graphical abstract can be visually improved if a possible bit cluttered. (not necessary)

Response: we agree with the reviewer and have improved the "cluttering" of the graphical abstract.

Minor corrections

The Acronym in the introduction

GWAS - genome-wide association study, not Whole-genome association studies

PSC- Primary sclerosing cholangitis, not sclerosing cholangitis

Pg 13 CRISPCas9 - was used, other places its CRISPCas

Figure 6b also in graphical abstract the colors used are very close and has no contrast - can be improved.

Fig 4-b the contrast can be improved if possible.

All minor corrections have been tackled. We thank the reviewer for his/her excellent review that improved our manuscript.